# Nutrient distribution and nitrogen and oxygen isotopic composition of nitrate in water masses of the subtropical South Indian Ocean

Natalie C. Harms[1], Niko Lahajnar[1], Birgit Gaye[1], Tim Rixen[1,2], Kirstin Dähnke[3], Markus Ankele[3], Ulrich Schwarz-Schampera[4] and Kay-Christian Emeis[1,3]

[1]Institute of Geology, Universität Hamburg, Hamburg, 20146, Germany.
[2]Leibniz-Centre for Tropical Marine Research, Bremen, 28359, Germany.
[3]Helmholtz-Zentrum Geesthacht (HZG), Institute for Coastal Research, Geesthacht, 21502, Germany.
[4]Federal Institute for Geosciences and Natural Resources (BGR), Hannover, 30655, Germany.

*Correspondence to*: Natalie C. Harms (natalie.harms@uni-hamburg.de)

**Abstract.** The Indian Ocean subtropical gyre (IOSG) is one of five extensive subtropical gyres in the world's ocean. In contrast to those of the Atlantic and Pacific Ocean, the IOSG has been sparsely studied. We investigate the water mass distributions based on T/S and oxygen data, and concentrations of water column nutrients and stable isotope composition of nitrate, using waters samples from two expeditions in 2016 (MSM 59/2) and 2017 (SO 259), collected between ~30°S and the equator. Our results are the first in this ocean region and provide new information on nitrogen sources and transformation

processes. We identify the thick layer of nutrient depleted surface waters of the oligotrophic IOSG with nitrate ($NO_3^-$) and phosphate ($PO_4^{3-}$) concentrations of <3 µmol/kg and <0.3 µmol/kg, respectively (<300 m; σ <26.4 kg/m³). Increased nutrient concentrations towards the equator represent the northern limb of the gyre, characterised by typical strong horizontal gradients of the outcropping nutriclines. The influx of the Subantarctic Mode Water (SAMW) from the Southern Ocean injects oxygen saturated waters with preformed nutrients, indicated by increased N and O isotope composition of nitrate

($\delta^{15}$N >7 ‰; $\delta^{18}$O >4 ‰) at 400−500 m (26.6−26.7 kg/m³), into the Subtropical thermocline. These values reflect partial N-assimilation in the Southern Ocean. Moreover, in the northern study area, a residue of nitrate affected by denitrification in the Arabian Sea is imported into intermediate and deep water masses (>27.0 kg/m³) of the gyre, indicated by an N deficit (N* ~−1 to −4 µmol/kg) and by elevated isotopic ratios of nitrate ($\delta^{15}$N >7 ‰; $\delta^{18}$O >3 ‰). Remineralisation of partial-assimilated organic matter, produced in the Subantarctic, leads to a decoupling of N and O isotopes in nitrate and results in

relatively low Δ(15-18) of <3 ‰ within the SAMW. In contrast, remineralisation of $^{15}$N-enriched organic matter originated in the Arabian Sea indicates higher Δ(15-18) values of >4 ‰ within the Red Sea-Persian Gulf Intermediate Water (RSPGIW). Thus, the subtropical South Indian Ocean is supplied by preformed nitrate from the lateral influx of water masses from regions exhibiting distinctly different N-cycle processes documented in the dual isotope composition of nitrate. Additionally, a significant contribution of $N_2$-fixation between 20.36° S−23.91° S is inferred from reduced $\delta^{15}$N-$NO_3^-$ values

towards surface waters (upward decrease of $\delta^{15}$N ~2.4 ‰), N* values of >2 µmol/kg and a relatively low Δ(15-18) of <3 ‰. A mass and isotope budget implies that at least 32-34 % of the nitrate in the upper ocean between 20.36° S−23.91° S is provided from newly fixed nitrogen, whereas $N_2$-fixation appears to be limited by iron or temperature south of 26° S.

## 1 Introduction

The South Indian Ocean is dominated by a subtropical anticyclonic gyre (Sarmiento and Gruber, 2006; Williams and Follows, 2003), the Indian Ocean Subtropical Gyre (IOSG), one of the major five subtropical gyres in the world's ocean. In contrast to those of the Atlantic and Pacific Oceans, where subtropical gyres occur north and south of the equator, the Indian Ocean developed only one subtropical gyre south of the equator. In comparison to other subtropical gyres, the IOSG has been sparsely investigated. Between 10−20° S, the South Equatorial Current marks the northern limb of the IOSG (SEC; Duing, 1970; Pickard and Emery, 1982; Woodberry et al., 1989) and separates the subtropical gyre of the South Indian Ocean from the southern equatorial Indian Ocean. In the centre of the subtropical gyre, Ekman transport leads to an intensive downwelling (Williams and Follows, 1998), which results in a deepening of thermo-, pycno- and nutriclines. These layers shoal towards the fringe of the IOSG causing steep horizontal gradients (McClain et al., 2004). Due to the intense downwelling and the resulting deepening of nutriclines, subtropical gyres form extensive oligotrophic regions, which occupy ~40 % of the Earth's surface (McClain et al., 2004). Since the biological productivity within these oligotrophic regions is relatively low they are often referred to as "oceanic deserts" (Clark et al., 2008). However, due to their immense size they contribute significantly to atmosphere-ocean carbon fluxes (McClain et al., 2004).

Future global warming is assumed to strengthen stratification in low-latitude oceans and to expand the low productive subtropical gyres, accompanied by a decrease of the net primary production (Behrenfeld et al., 2006). This might have crucial impact on the marine nitrogen cycle. To study the marine nitrogen cycle, we use nitrate and phosphate concentrations as well as the isotopic signature of nitrate (Deutsch et al., 2001; Deutsch et al., 2007; Gruber and Sarmiento, 1997; Lehmann et al., 2005; Sigman et al., 2005). The dominant source and sink of fixed, reactive nitrogen in the ocean are diazotrophic $N_2$-fixation and heterotrophic denitrification (Deutsch et al., 2001). $N_2$-fixation by diazotrophs, such as *Trichodesmium* is observed over much of the tropical and oligotrophic subtropical oceans (Karl et al., 1995; Michaels et al., 1996; Capone et al., 1997; Emerson et al., 2001). $N_2$-fixation compensates the loss of reactive nitrogen during the heterotrophic denitrification if the ocean's marine nitrogen cycle is in a steady state (Deutsch et al., 2001).

The inputs of nitrogen (N) through $N_2$-fixation are detached from inputs of phosphorus (P), leading to a decoupling of the nitrate ($NO_3^-$) and phosphate ($PO_4^{3-}$) pool. Deviations in the $NO_3^-$ to $PO_4^{3-}$ relationship from the Redfield-stoichiometry are used to study rates of both, $N_2$-fixation and denitrification (Gruber and Sarmiento, 1997; Sigman et al., 2005). Therefore, the tracer N* is used as an indicator for excesses and deficits in $NO_3^-$ relative to the global $NO_3^-/PO_4^{3-}$ ratio and is expressed by the formula $N*=[NO_3^-]–16*[PO_4^{3-}]+2.9$ µmol/kg. The concept of N* has been discussed in detail by Gruber and Sarmiento (1997) and slightly modified by Deutsch et al. (2001). The concentration of 2.9 µmol/kg was added to bring the global mean of N* to about zero (Sarmiento and Gruber, 2006). However, the use of N* has limitations. First, the deviation from the Redfield-stoichiometry may not always be a result of N inputs or outputs ($N_2$-fixation and denitrification) but may reflect also variations of uptake and remineralisation processes (Sigman et al., 2005). Second, input and losses partially overprint each other when they occur simultaneously in the same water body.

We use stable isotopes of nitrate (N and O) to overcome the weakness associated with the N* approach and distinguish between sources and sinks of fixed nitrogen to study transfer processes in the nitrogen cycle (e.g., N-assimilation, denitrification, nitrification, $N_2$-fixation), also when they occur simultaneously. Isotope ratios are reported in ‰ using the δ-notation ($\delta^{15}N = [(^{15}N/^{14}N_{sample})/(^{15}N/^{14}N_{atm.N2})]–1*1000$; $\delta^{18}O = [(^{18}O/^{16}O_{sample})/(^{18}O/^{16}O_{VSMOW})]–1*1000$, with air $N_2$ and

VSMOW as reference for $^{15}N/^{14}N$ and $^{18}O/^{16}O$, respectively). During consumption processes of nitrate, e.g., N-assimilation or denitrification, lighter isotopes are preferentially assimilated, leaving the substrate enriched in $^{15}N$ and $^{18}O$ according to its isotope effect ($^{15}\varepsilon$ and $^{18}\varepsilon$, e.g., $^{15}\varepsilon$ is defined as $^{14}k/^{15}k–1$, where $^{14}k$ and $^{15}k$ are the rate coefficients of the reactions for the $^{14}N$- and $^{15}N$-bearing forms of nitrate). Several culture experiments indicate that $\delta^{15}N$ and $\delta^{18}O$ of the residual nitrate pool rise equally as consumption proceeds, consequently the O-to-N isotope effect ($^{18}\varepsilon$:$^{15}\varepsilon$) is close to 1 (Granger et al., 2004;

Rafter et al., 2013; Sigman et al., 2003; Sigman et al., 2005).

While nitrate consumption processes such as N-assimilation and denitrification lead to indistinguishable imprints on N and O isotope compositions, nitrate production processes (nitrification and $N_2$-fixation) have very different effects on the N and O isotopes of nitrate (Rafter et al., 2013; Sigman et al., 2005). Whereas almost all of the ammonium generated from organic N is oxidized to nitrate in oxic subsurface waters of the open ocean, the N isotope effect associated with ammonium production

and nitrification do not affect the $\delta^{15}N$-$NO_3^-$. Therefore, the N isotope effect depends more on the biomass being remineralised (Rafter et al., 2013; Sigman et al., 2005). In contrast, the $\delta^{18}O$ of newly nitrified nitrate is independent of the isotopic composition of the organic matter and leads to a counteracting behaviour of $\delta^{15}N$ and $\delta^{18}O$. Consequently, the decoupling of N and O isotopes provide a better understanding of nitrate assimilation and regeneration processes in marine environments (Casciotti et al., 2008; DiFiore et al., 2009; Sigman et al., 2005, 2009, Wankel et al., 2007).

Our investigations in the South Indian Ocean are part of environmental studies in the INDEX (Indian Ocean Exploration) program for marine resource exploration by the federal Institute for Geosciences and Natural Resources (BGR), Germany, and the International Seabed Authority (ISA). We use CTD measurements and analyse seawater samples to determine nutrient concentrations and stable isotopes of nitrate ($\delta^{15}N$ and $\delta^{18}O$) along a transect from the IOSG to the southern equatorial Indian Ocean. The main goal of this study is to investigate the relatively unknown hydrology and the unexplored

distribution of nutrients and stable isotopes of nitrate to identify N-cycle processes within the IOSG towards the equatorial South Indian Ocean. First, we identify the water masses and their provenance by their unique characteristic physical properties and establish the first water mass distribution model for this ocean region. In a second step, we use new nutrient and stable isotope data to determine nutrient sources to the IOSG and their role in the marine nitrogen cycle. Furthermore, we demonstrate the influence of water masses on the nutrient distribution and the isotopic composition of water column

nitrate by the influx of preformed nutrients. Our results of nutrient and isotope measurements are the first in the IOSG and bridge the gap between several investigations in the Arabian Sea (e.g., Brandes et al., 1998; Gaye-Haake et al., 2005; Gaye et al., 2013; Ward et al., 2009) and in the Indian section of the Southern Ocean (e.g., Bianchi et al., 1997; DiFiore et al., 2006; DiFiore et al., 2010; Sigman et al., 1999, 2000).

## 2 Materials and Methods

### 2.1 CTD measurement and sample collection

In total, 313 seawater samples were collected at 15 CTD stations (Fig. 1) during two expeditions with R/V *Maria S. Merian* (MSM 59/2 "INDEX 2016-2"; November−December 2016) and R/V *Sonne* (SO 259 "INDEX 2017"; August−October

2017). The CTD was equipped with sensors to determine density, temperature, salinity and oxygen at overall 17 CTD stations from the surface down to the sea floor. No water samples were collected at stations 07 and 11.

The study area covers the region of the IOSG from 30°S, across the SEC at 10−20°S and towards the equator. Fourteen CTD stations are located within the IOSG from 20.36° S to 27.78° S and 67.07° E to 73.92° E. CTD 05 is located in the region of the SEC (15.08° S, 74.05° E) and at the northern end of the IOSG. The northernmost CTD stations (CTD 01, 03; 2017) at

8.81°S 75.67°E and 2.98°S 77.16°E are positioned in the southern equatorial Indian Ocean, north of the SEC. Seawater samples were collected for measurements of nutrients and stable isotopes of nitrate. Samples were filtered through a Nucleopore polycarbonate filter (0.4 μm) with a metal- and silicon-free Nalgene filtration unit. The filtered water was bottled in Falcon PE tubes (45 ml) and immediately stored at −20°C during the cruise. The samples were shipped as frozen airfreight (−20°C) to Germany. Nutrient concentrations and stable isotopes of nitrate (N and O) were determined in the home lab

immediately after arrival.

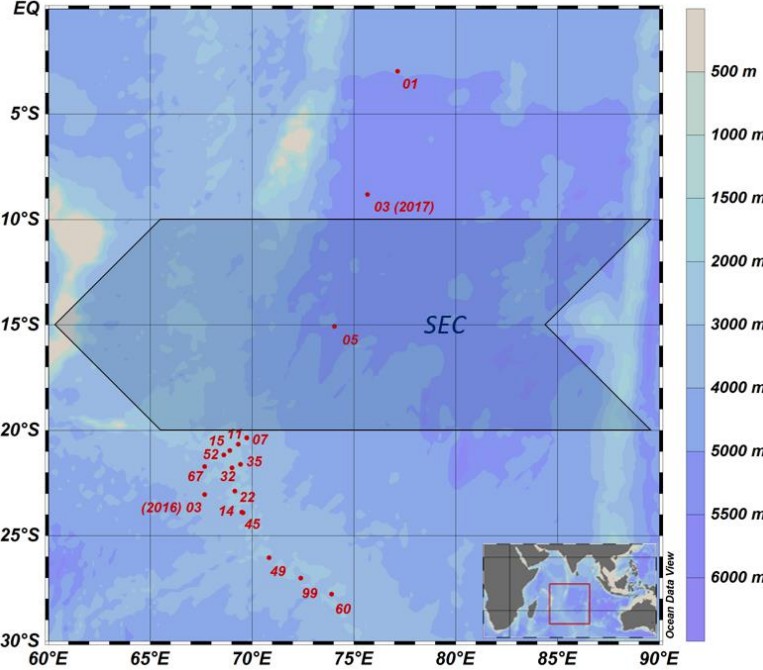

**Figure 1:** Sampling location during the cruises MSM 59/2 (INDEX 2016-2) and SO 259 (INDEX 2017). Shaded arrow represents the westward-directed, broad South Equatorial Current (SEC) after Woodberry (1989) from 10−20°S. Colours denote water depths.

## 2.2 Nutrient analyses

Nutrient concentrations ($NO_x$, $NO_2^-$, $NH_4^+$, $PO_4^{3-}$) were measured with a SEAL AutoAnalyzer3HR with standard colourimetric methods (Grasshoff et al., 2009). Ammonia and nitrite concentrations were below detection limit. Nitrate determination included reduction of nitrate to nitrite with a cadmium reduction column. Nitrite ions reacted with sulphanilamide to form a diazo compound, followed by a reaction to an azo dye with N-(1-naphtyl-)ethylenediamine (NEDD) and was measured at 520−560 nm. Phosphate determination followed the method of Murphy and Riley (1962). Under acid conditions a phosphomolybdic complex was formed of ortho- phosphate, antimony and molybdate ion (Wurl, 2009). Followed by reduction of ascorbic acid, the blue colour complex was measured at 880 nm. The relative error of duplicate sample measurements was below 1.5 % for nitrate and phosphate concentrations and detection limit was <0.5 µmol/kg for $NO_x$, and >0.1 µmol/kg for $PO_4^{3-}$.

## 2.3 Measurements of N and O isotopes of nitrate

Isotope measurements were only conducted for samples with nitrate concentrations >1.7 µmol/kg. Stable isotopes of nitrate ($\delta^{15}N$ and $\delta^{18}O$) were determined using the "denitrifier" method (Casciotti et al., 2002; Sigman and Casciotti, 2001). Nitrate was converted to $N_2O$ gas using denitrifying bacteria (*Pseudomonas aureofaciens*). Based on nitrate concentrations, sample volumes were adjusted to yield 10 nmoles $N_2O$ and were injected into suspensions of *Pseudomonas aureofaciens* (ATCC#13985) for combined analysis of $\delta^{15}N$ and $\delta^{18}O$. The resulting $N_2O$ gas in headspace was purged into a GasBench II (ThermoFinnigan), and analysed in a Delta V Advantage and a Delta V Plus mass spectrometer. The results were calibrated using IAEA-N3 ($\delta^{15}N\text{-}NO_3^- = +4.7$ ‰ and $\delta^{18}O\text{-}NO_3^- = +25.6$ ‰) and USGS-34 ($\delta^{15}N\text{-}NO_3^- = -1.8$ ‰ and $\delta^{18}O\text{-}NO_3^- = -27.9$ ‰) (Böhlke et al., 2003). A further internal potassium nitrate standard (KBI) was analysed within each run for quality assurance ($\delta^{15}N\text{-}NO_3^- = +7.1$ ‰). Isotope values were corrected using the "bracketing scheme" from Sigman et al. (2009) for $\delta^{18}O\text{-}NO_3^-$ and a two-point correction referred to IAEA-N3 and USGS-34 for $\delta^{15}N\text{-}NO_3^-$ and $\delta^{18}O\text{-}NO_3^-$. The standard deviation for IAEA-N3 was better than 0.2 ‰ for $\delta^{15}N\text{-}NO_3^-$ and 0.3 ‰ for $\delta^{18}O\text{-}NO_3^-$, which is within the same specification for $\delta^{15}N\text{-}NO_3^-$ and $\delta^{18}O\text{-}NO_3^-$ for at least duplicate measurements of the samples.

# 3 Results

## 3.1 Physical water column properties

South of 25° S the upper 170 m are characterised by an intense salinity maximum with values of >35.5 PSU and temperatures above 15°C (Figs. 2a, 3). The salinity maximum is carried northwards and is subducted underneath the surface layer within a temperature range of 22−15°C and with a core density of σ=25.5 kg/m³ (~250 m). Further north (CTD 03, 2017; 8.81° S) at the same density level, the salinity is significantly lower and reveals values of 35.2 PSU. The northernmost station (CTD 01, 2017; 2.89°S) indicates again a slight increase in salinity (>35.3 PSU). Between 22° S and 10° S, less

saline surface water (<35.1 PSU) lies above the density level of the salinity maximum with temperatures of >23°C and densities above 24.0 kg/m³ (<150 m). South of 15° S, directly underneath the salinity maximum an oxygen maximum with values of >4.7 ml/l occurs at a density range of 26.4−26.9 kg/m³ (250−750 m; Fig. 2b) and temperatures between 8°C and 15°C (Fig. 3). The lower limit of the oxygen maximum coincides with a temperature level of 8−9°C at σ=26.9−27.0 kg/m³ and marks the permanent thermocline at a depth of ~750 m in the south and at a depth of ~500 m in the north. Oxygen concentrations decrease towards the north and fall below 2 ml/l at the northernmost stations (CTD 01, CTD 03; 2017; Figs. 2b, 3). Below the permanent thermocline (<9°C), an absolute salinity minimum with values less than 34.6 PSU is found in the southern region (Figs. 2a, 3), within a density range of 26.9−27.4 kg/m³ (core density σ=27.2 kg/m³), which is strongly diluted further north and temperatures are below 8°C (Fig. 3). In the southern equatorial Indian Ocean at CTD 01, an increase in salinity (>34.9 PSU; σ=27.2 kg/m³) corresponds with reduced oxygen concentrations of <1.1 ml/l. Overall, low oxygen concentrations dominate the northern study area and extend to deeper water masses at the southernmost stations (<3.5 ml/l; Figs. 2b, 3).

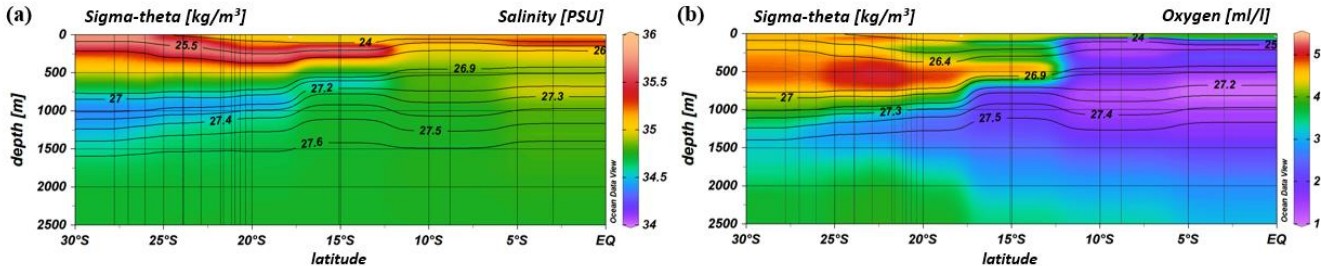

**Figure 2:** Profiles of salinity (a) and oxygen distribution (b) from CTD measurements during cruises MSM 59/2 (2016) and SO 259 (2017). Contour lines indicate the potential density sigma-theta in kg/m³.

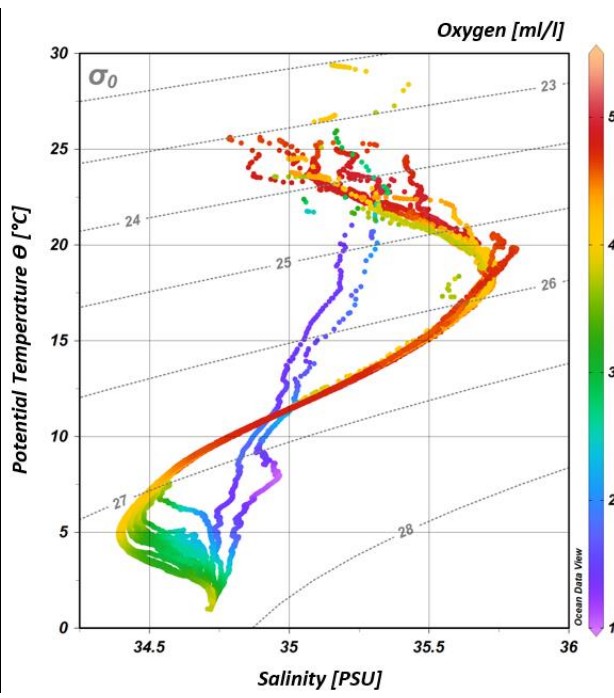

**Figure 3:** T-S diagram (potential temperature vs. salinity) from CTD measurements during cruises MSM 59/2 (2016) and SO 259 (2017). The colour bar indicates oxygen concentrations and grey, dotted lines represent density surfaces in sigma-theta (kg/m³). The northernmost CTD stations (CTD 01, CTD 03; 2017) are characterized by low oxygen concentrations (<2 ml/l) and less variations in the salinity distribution, while the water column profiles in the IOSG indicate a distinct salinity maximum and minimum, respectively.

### 3.2 Nutrient concentration

Within the subtropical gyre, the upper 100 m are strongly depleted in nitrate and phosphate with concentrations less than 1 µmol/kg of nitrate and less than 0.1 µmol/kg of phosphate (Fig. 4a, 4b; Table 1). Within the depth range of the salinity maximum (24.9−26.4 kg/m³; <300 m), nutrient concentrations are still minor with $NO_3^-$ and $PO_4^{3-}$ values of <3 µmol/kg and <0.3 µmol/kg, respectively (Table 1). Nutrient concentrations rise within the depth range of the oxygen maximum ($\sigma$= 26.4−26.9 kg/m³), where we observe concentrations of ~11 µmol/kg $NO_3^-$ and <0.9 µmol/kg $PO_4^{3-}$ before they reach typical deep-sea values of >30 µmol/kg and >2 µmol/kg (Sarmiento and Gruber, 2006) within intermediate waters (>26.9 kg/m³; >750 m). Across the northern fringe of the gyre at surface waters, (CTD 05, 2017; 15.08° S) nutrient concentrations slightly increase (Figs. 4a, 4b).

Further north, at stations CTD 01 and 03 (2.98−8.81° S), nutrient concentrations in the upper water column reach values typical for open ocean areas that are unaffected by gyral downwelling or high biological production (Figs. 4a, 4b). At 90 m water depth, concentrations were ~11 µmol/kg for nitrate and ~1 µmol/kg for phosphate. Within the thermocline (23.0−27.0 kg/m³; <550 m), nutrient concentrations attain values of >20 µmol/kg for nitrate and >1.5 µmol/kg for phosphate, before they level out at values of >35 µmol/kg for nitrate and >2.5 µmol/kg for phosphate at greater depth (Table 1).

### 3.3 N and O isotopes of nitrate

In the upper 750 m (<26.9 kg/m³), distinct N and O isotope maxima with $\delta^{15}N$ of >7.0 ‰ and $\delta^{18}O$ of >4.0 ‰ are found at latitudes 27.78° S–15.08° S (Table 1; Figs. 4c, 4d). N and O isotope maxima are observed at ~400−500 m (26.6−26.7 kg/m³) and correlate with the oxygen maximum of >4.7 ml/l. At latitudes 23.91° S−20.96° S, the N isotope maximum is found at 400 m, whereas the O isotope maximum is observed at 500 m. Consequently, N and O isotope maxima indicates an offset of ~100 m (see supplement tables S2 and S3). Above the isotopic maxima, both $\delta^{15}N$ and $\delta^{18}O$ decrease to values of ~5.4 ‰ and ~2.1 ‰, respectively, in the upper 300 m; an exception are the southernmost stations (CTD 49, 60, 99; 2017), where elevated $\delta^{15}N$ values extend up to the surface (Fig. 4c). In surface waters further north (<250 m), $\delta^{15}N$ and $\delta^{18}O$ increase to values of >7.0 ‰ and >4 ‰, respectively, at the northernmost station (CTD 01, 2017; Figs. 4c, 4d). Underneath this surface layer, N and O isotope ratios slightly decrease at ~180 m, before $\delta^{15}N$ and $\delta^{18}O$ again rise to >7.0 ‰ and >3.0 ‰, with an extended maximum in the depth interval from 300 m to 900 m (<27.3 kg/m³) that coincides with elevated salinities. Below the isotopic maxima in the southern region at ~400−500 m and below the depth interval with high δ-values in the northernmost CTD station, $\delta^{15}N$ and $\delta^{18}O$ decrease towards deeper waters and have average values of 5.8 ‰ and 2.3 ‰ (Figs. 4c, 4d).

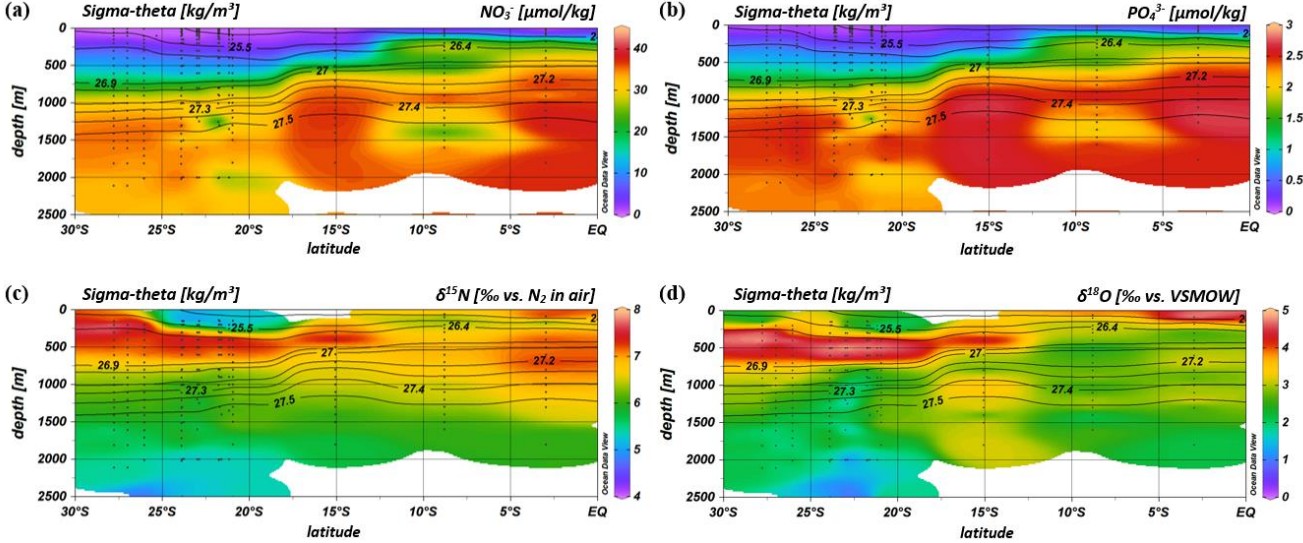

**Figure 4:** Profiles of nitrate (a) and phosphate concentrations (b) and $\delta^{15}N$-NO$_3^-$ (c) and $\delta^{18}O$-NO$_3^-$ (d) of seawater samples collected during cruises MSM 59/2 (2016) and SO 259 (2017). Contour lines indicate the potential density sigma-theta in kg/m³.

**Table 1:** Average nitrate and phosphate concentrations and average $\delta^{15}$N-NO$_3^-$ and $\delta^{18}$O-NO$_3^-$ values within water masses of the South Indian Ocean, defined along their potential density surfaces (sigma-theta) and separated into four latitudinal sections (27.78–26.05 °S; 23.91–20.36 °S; 15.08 °S; 8.81–2.98 °S). Water mass abbreviations as followed: Indonesian Upper Water (IUW), Subtropical Surface Water (SSW), Subantarctic Mode Water (SAMW), Indian Equatorial Water (IEW), Antarctic Intermediate Water (AAIW), Indonesian Intermediate Water (IIW), Red Sea-Persian Gulf Intermediate Water (RSPGIW), Indian Deep Water (IDW), and Circumpolar Deep Water (CDW).

| Latitude [°S] | Sigma-theta [kg/m³] | Water masses | NO$_3^-$ [µmol/kg ± 1 D.A.] | PO$_4^{3-}$ [µmol/kg ± 1 D.A.] | $\delta^{15}$N-NO$_3^-$ [‰ ± 1 D.A.][a] | $\delta^{18}$O-NO$_3^-$ [‰ ± 1 D.A.][b] |
|---|---|---|---|---|---|---|
| 27.78–26.05 | <26.4 | SSW | 1.67 ± 1.59 | 0.24 ± 0.15 | 7.62 ± 0.30 | 2.94 ± 0.47 |
| | 26.4–26.9 | SAMW | 11.43 ± 4.01 | 0.88 ± 0.24 | 7.25 ± 0.31 | 4.34 ± 0.52 |
| | 26.9–27.4 | AAIW | 28.71 ± 5.24 | 2.05 ± 0.32 | 6.28 ± 0.16 | 2.95 ± 0.40 |
| | >27.4 | Deep Water Masses | 34.55 ± 1.02 | 2.48 ± 0.07 | 5.68 ± 0.29 | 2.17 ± 0.26 |
| 23.91–20.36 | <24.9 | Surface Water/IUW | 0.37 ± 0.26 | 0.08 ± 0.03 | N/D | N/D |
| | 24.9–26.4 | SSW | 2.85 ± 0.98 | 0.29 ± 0.08 | 5.41 ± 0.54 | 2.07 ± 0.44 |
| | 26.4–26.9 | SAMW | 11.04 ± 2.76 | 0.82 ± 0.17 | 7.35 ± 0.25 | 4.60 ± 0.29 |
| | 26.9–27.4 | AAIW | 28.57 ± 3.23 | 1.99 ± 0.23 | 6.22 ± 0.13 | 2.21 ± 0.32 |
| | >27.4 | Deep Water Masses | 32.72 ± 2.54 | 2.34 ± 0.16 | 5.45 ± 0.30 | 1.62 ± 0.32 |
| 15.08 | <24.3 | IUW | 1.69 ± 1.15 | 0.30 ± 0.06 | N/D | N/D |
| | 24.3–26.6 | SSW | 4.42 ± 3.27 | 0.37 ± 0.15 | N/D | N/D |
| | 26.6–26.9 | SAMW | 15.19 ± 2.70 | 1.11 ± 0.16 | 7.19 ± 0.26 | 4.01 ± 0.40 |
| | 26.9–27.2 | AAIW/IIW | 30.74 ± 3.30 | 2.18 ± 0.26 | 6.68 ± 0.05 | 2.94 ± 0.08 |
| | >27.2 | Deep Water Masses | 36.28 ± 0.89 | 2.58 ± 0.15 | 6.06 ± 0.49 | 3.03 ± 0.45 |
| 8.81–2.98 | <23.0 | Surface Water | 0.61 ± 0.26 | 0.17 ± 0.04 | 6.66 ± 0.41 | 3.77 ± 0.55 |
| | 23.0–27.0 | IEW | 22.84 ± 5.85 | 1.57 ± 0.34 | 6.96 ± 0.07 | 2.86 ± 0.14 |
| | 27.0–27.3 | RSPGIW | 32.77 ± 3.23 | 2.35 ± 0.19 | 7.03 ± 0.09 | 2.85 ± 0.23 |
| | 27.3–27.7 | IDW | 34.21 ± 4.59 | 2.48 ± 0.23 | 6.56 ± 0.19 | 2.72 ± 0.36 |
| | >27.7 | CDW | 35.92 ± 0.48 | 2.41 ± 0.05 | 5.32 ± 0.24 | 2.07 ± 0.13 |

[a] $\delta^{15}$N-NO$_3^-$ in ‰ versus air and [b] $\delta^{18}$O-NO$_3^-$ in ‰ versus VSMOW; D.A. = deviation from the average value; N/D = not detectable due to insufficient nitrate concentrations.

## 4 Discussion

### 4.1 Water mass distribution

Water masses in the study area are well discernible by their densities, salinities and oxygen concentrations (Fig. 5). In accordance with definitions from the literature, we identified water masses from the IOSG towards the southern equatorial Indian Ocean and established the first water mass distribution model for this ocean region (Fig. 6). To generate the water mass distribution model, we use salinity and oxygen distributions along sigma-theta surfaces. We separate our study area in three latitudinal sections, which demonstrate the alteration of water masses along the latitudinal transect and between the different ocean regimes (Fig. 5). We present the provenance of water masses of Antarctic and Subantarctic origin converging and mixing with water masses from the southern equatorial Indian Ocean and the Arabian Sea. The water mass distribution model serves as a basis for the understanding of our nutrient and coupled N and O isotope measurements of nitrate.

### 4.1.1 Surface and thermocline waters (<26.9 kg/m³; <800 m)

A high salinity surface layer (>35.5 PSU) centred at ~25.5 kg/m³ (Fig. 5a) is described in several studies. It has been termed "southern subtropical surface water" by Muromtsev (1959), "subtropical surface water" by Wyrtki (1973) and "subtropical subsurface water" (SSW) by Schott and McCreary (2001). For further descriptions, we adopt the definition of Wyrtki (1973) and use the abbreviation SSW. The SSW is formed in the subtropical gyre of the southern hemisphere by excess of evaporation over precipitation (Schott and McCreary, 2001) at latitudes 25−35° S (Baumgartner and Reichel, 1975). It is subducted into the thermocline of the subtropical gyre (Schott and McCreary, 2001), is detectable as far north as 15.08° S at CTD 05 (Fig. 3b) and not discernable further north in the southern equatorial Indian Ocean (Figs. 5c, 6).

Less saline surface water (<35.1 PSU) occurs above the density level of the salinity maximum (>23°C; <24.0 kg/m³; Fig. 5b) and is described by Wyrtki (1971) and Warren (1981). These low salinity values reflect an excess of precipitation over evaporation at latitudes 0−10° S (Baumgartner and Reichel, 1975) accompanied by the influx of low salinity water (34.0−34.5 PSU) from the Pacific Ocean through the Indonesian Archipelago, called "Indonesian Throughflow" (ITF). The ITF carries less saline water westwards by the SEC within the entire thermocline (Wyrtki, 1971; You and Tomczak, 1993). Emery (2001) named this less saline surface water (34.4−35.0 PSU) "Indonesian Upper Water" (IUW; Fig. 6).

The oxygen maximum south of 20° S in a density range of 26.4−26.9 kg/m³ (250−750 m; Fig. 5d) corresponds with the "Subantarctic Mode Water" (SAMW; Figs. 5, 6), described by McCartney (1977). It is formed at latitudes 40° S−50° S and injects oxygen saturated waters at a temperature range of 6−14°C into the subtropical gyre. The SAMW in the South Indian Ocean can be separated into three modes by slightly different density distributions (Herraiz-Borreguero and Rintoul, 2011) which are originated in different ocean regions. For example, a lighter mode of the SAMW is formed in the western Indian basins and is limited to the southwest portion of the subtropical gyre, while the denser mode is found south off Australia and is carried further north by the outer portion of the subtropical gyre and ventilate a larger fraction of the gyre interior (Herraiz-Borreguero and Rintoul, 2011). However, for our purposes we assume the SAMW as one homogenous water mass flowing above the density surface of 26.9 kg/m³. On its transition to the north, the oxygen concentrations rapidly decrease from >4.6 ml/l (CTD 05; Fig. 5e) to <1.9 ml/l (CTD 01, 03; Fig. 5f) because of respiration and the absence of effective ventilation in the northern Indian Ocean. The reduced vertical changes in salinity north of ~15°S mark the "Indian Equatorial Water" (IEW; Fig. 6). This is described by Sharma (1976), Warren (1981), Quadfasel and Schott (1982), You and Tomczak (1993) and Schott and McCreary (2001) as a mixture of thermocline water masses from the northern and southern Indian Ocean.

### 4.1.2 Intermediate water masses (26.9−27.4 kg/m³; 800−1000 m)

The salinity minimum (<34.6 PSU) south of 15° S, in a density range of 26.9−27.4 kg/m³ (core density σ=27.2 kg/m³; Fig. 5a) is associated with the "Antarctic Intermediate Water" (AAIW; Fig.4; Bindoff and McDougall, 2000; Deacon, 1933; Fine, 1993; Schott and McCreary, 2001; Toole and Warren, 1993; Warren, 1981; Wyrtki, 1973; You, 1998). It is transported eastwards by the "Antarctic Circumpolar Current" (ACC), penetrates into all three oceans and extends towards the equator to

feed the intermediate waters (Fine, 1993; McCartney, 1977; Piola and Gordon, 1989; Reid, 1986, 1989; Sverdrup et al., 1942; Talley, 1996; Wüst, 1935).

The salinity minimum (<34.6 PSU) observed at station CTD 05 (15.08° S; Fig. 5b) has a slightly divergent core density (27.0 kg/m³) compared to the AAIW (Fig. 5a). This implies a further source to the salinity minimum of the AAIW. A low
salinity water mass (~34.8 PSU) flows along 10−15° S (Schott and McCreary, 2001; Wyrtki, 1971; You and Tomczak, 1993) and originates from the ITF. At intermediate depths it has been called "Indonesian Intermediate Water" (IIW; Fig. 6) by Emery and Meincke (1986) and Emery (2001).

The increase in salinity (>34.9 PSU; Fig. 5c) further north, at the same density level as the AAIW, is caused by the inflow of saline water from the Arabian Sea, mainly from the Red Sea outflow (Warren, 1981) and is additionally fed by the outflow
of the Persian Gulf (Emery and Meincke, 1986). Therefore, this water mass is called "Red Sea-Persian Gulf Intermediate Water" (RSPGIW; Fig. 6). The RSPGIW is transported towards the equator and beyond to as far south as 10° S (You, 1998), recirculates in the tropical gyre, and creates the absolute oxygen minimum (<1.1 ml/l) caused by biogeochemical processes in the Arabian Sea (see Sect. 4.2.1).

### 4.1.3 Deep water masses (>27.4 kg/m³; >1000 m)

Overall, low oxygen concentrations in the northern study area underneath the AAIW (>27.4 kg/m³; Fig. 5f) are caused by in situ consumption (Wyrtki, 1962) and reduced ventilation in the northern Indian Ocean. The deep oxygen minimum extends towards the south (~3.0 ml/l) and is associated with the water mass of the "Indian Deep Water" (IDW). The IDW has higher salinities than the overlying AAIW (Bindoff and McDougall, 2000; Mantyla and Reid, 1995; Schott and McCreary, 2001; Talley, 2013) with values of >34.6 PSU below the density range of the AAIW (Fig. 5a). The IDW (σ=~27.5 kg/m³) flows in
the density range just above the "Circumpolar Deep Water" (CDW; Fig. 6) and a further increase in salinity (34.62−34.73 PSU) and in the oxygen concentration at the 2°C temperature level (Emery, 2001) mark the transition between the IDW and the underlying CDW.

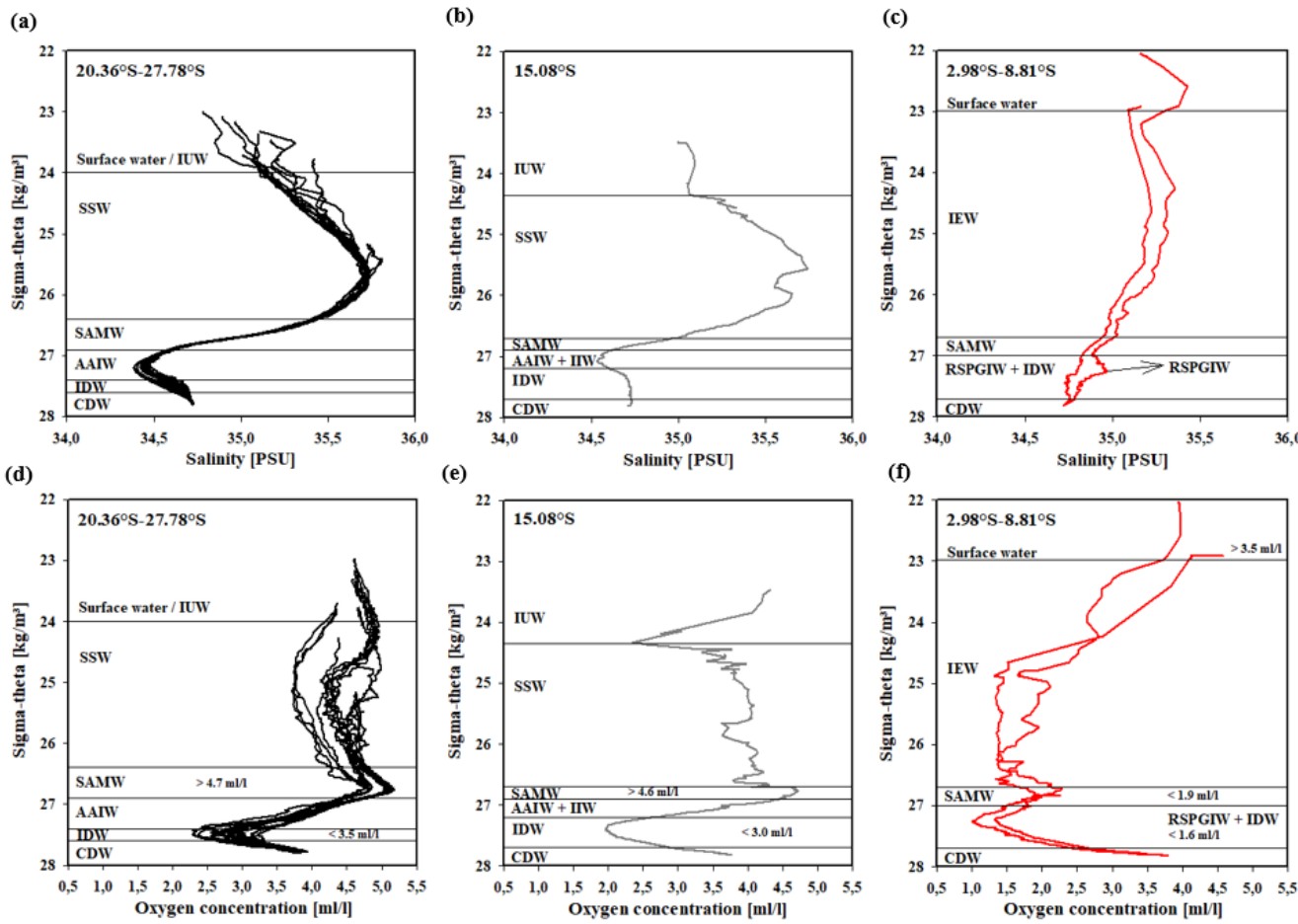

**Figure 5:** Water mass properties represented as salinity vs. sigma-theta diagrams (a, b, c) and as oxygen vs. sigma-theta diagrams (d, e, f) for CTD stations at latitudes 20.36° S−27.78° S, 15.08° S and 2.98° S−8.81° S. For water mass abbreviations see Table 1.

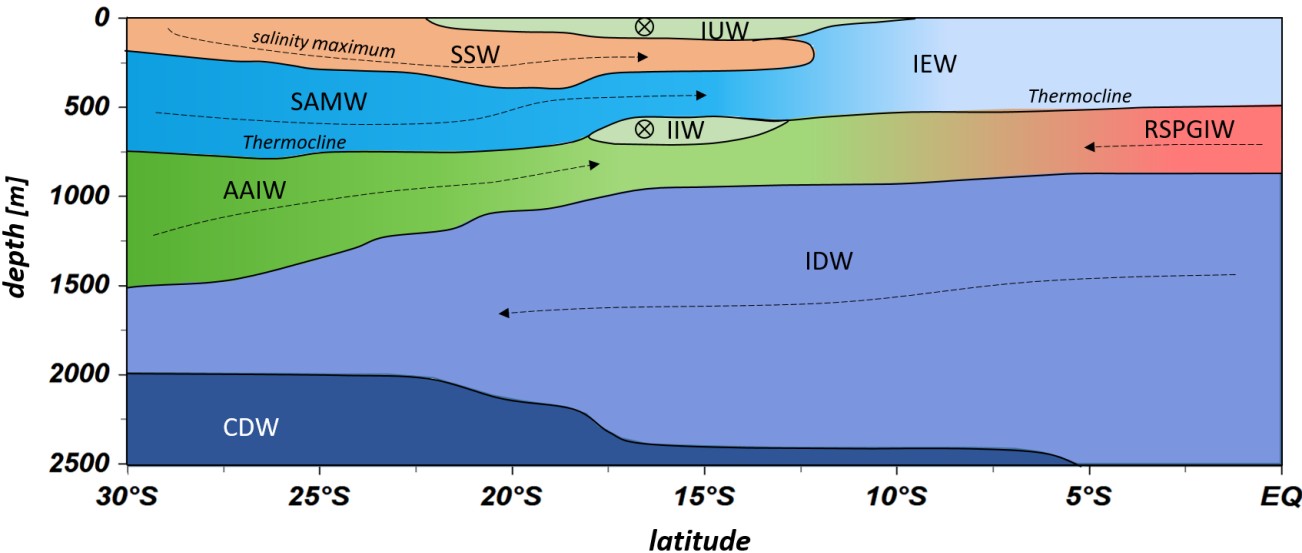

**Figure 6:** Water mass distribution model from 30° S to the equator. The CTD stations taken into account for this transect ranged between 67.07 °E and 77.16 °E. Dotted lines represent N-S current directions and circled crosses indicate latitudinal directions from E to W. For water mass abbreviations see Table 1.

## 4.2 Nutrient distribution and N-cycle processes

### 4.2.1 Nutrient supply in the oligotrophic subtropical gyre and lateral transfer across the gyre boundaries

Intense downwelling in the centre of the IOSG is induced by the convergence of horizontal Ekman volume flux (Williams and Follows, 2003) and creates the thick layer of nutrient depleted surface waters within the IUW and SSW (Fig. 7a, 7b, Table 1), and also within the underlying SAMW. The northward increase in nutrients at ~15° S (CTD 05, 2017) marks the northern boundary of the subtropical gyre and the maximum extension of the IUW, SSW, and SAMW (Table 1). Further increase in nutrient concentrations within the IEW indicate the transition from the subtropical gyre towards the southern equatorial Indian Ocean identified by the characteristic shoaling of the nutricline at the northern fringe of the gyre (Table 1, Fig. 7a, 7b). The IEW is not a well-defined water mass, but rather a mixture of thermocline waters from the South Indian Ocean and from the nutrient-enriched northern Indian Ocean. Therefore, just below the upper 100 m nutrient concentrations increase up to ~23 µmol/kg nitrate and ~1.6 µmol/kg phosphate (Table 1) at the northernmost stations (CTD 01, 03; 2017) and indicate the increasing influence of the nutrient-enriched northern Indian Ocean (Gaye et al., 2013). This increased northern influence is also reflected by the $NO_3^-/PO_4^{3-}$ ratios, which exhibit values of less than 8 in the upper 200 m of the subtropical gyre, but increase towards the southern equatorial Indian Ocean, tracking the outcropping nutriclines (Fig. 7c). Low $NO_3^-/PO_4^{3-}$ ratios are typical in surface waters of oligotrophic regions because nitrate commonly becomes depleted prior to phosphate (Sarmiento and Gruber, 2006; Deutsch et al., 2007). Due to the intense downwelling in the centre of the IOSG, the supply of nutrients by vertical mixing is reduced or absent in the gyre (Williams and Follows, 1998). Thus, lateral

transfer across the gyre boundaries and biologically N₂-fixation are major processes supplying nutrients to the euphotic zone of the subtropical gyre.

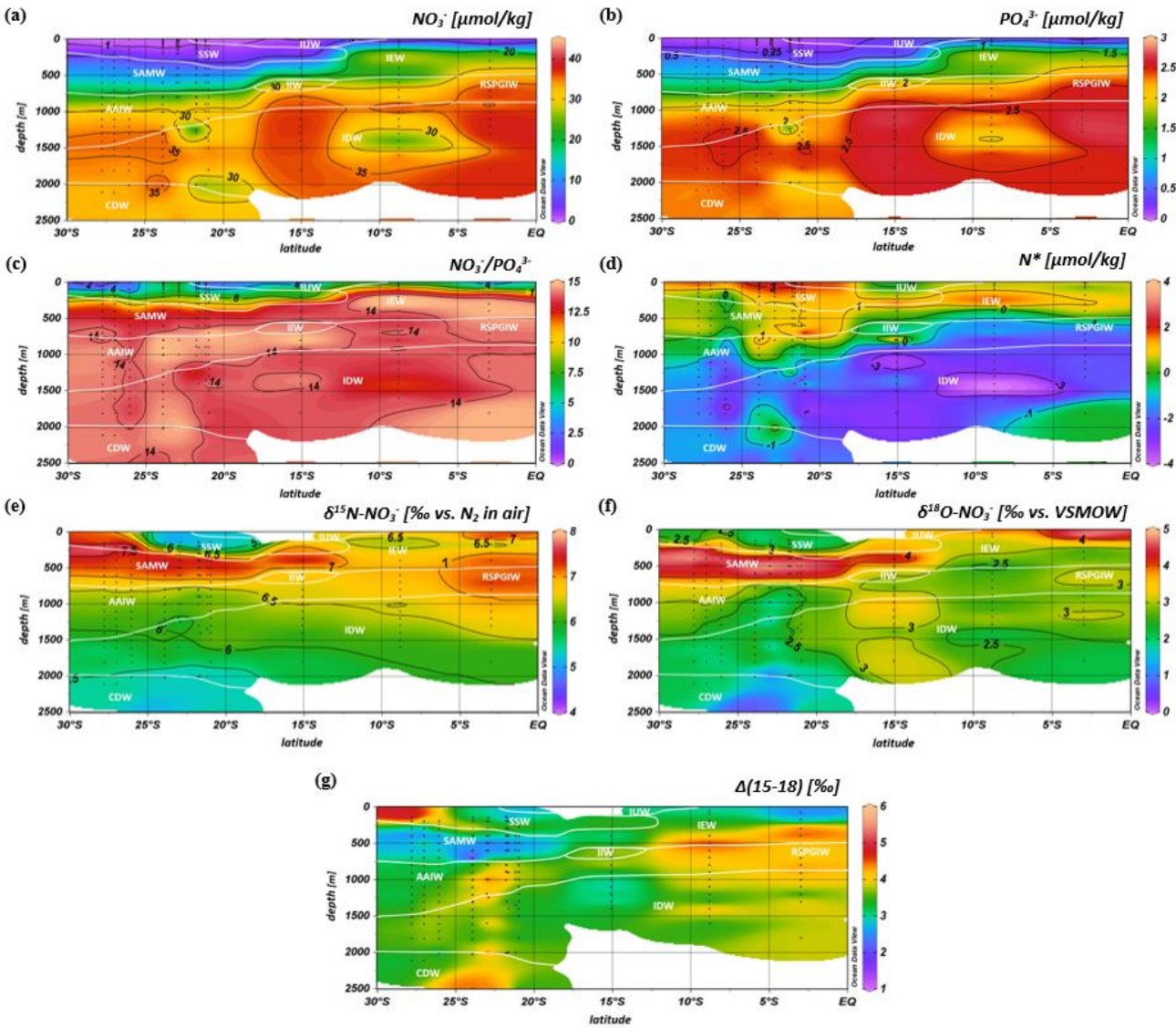

**Figure 7:** Latitudinal profiles from south to north with an overlay of the water mass distribution model (white contour lines) in the South Indian Ocean of nitrate (a) and phosphate concentrations (b), $NO_3^-/PO_4^{3-}$ ratio (c), N* (d), $\delta^{15}N\text{-}NO_3^-$ (e) and $\delta^{18}O\text{-}NO_3^-$ of nitrate (f) and nitrate $\Delta(15\text{-}18)$ as the difference between $\delta^{15}N\text{-}NO_3^-$ and $\delta^{18}O\text{-}NO_3^-$. For water mass abbreviations see Fig. 6.

The water masses entering the study area from the Southern Ocean and from the northern Indian Ocean have characteristic nutrient concentrations and isotope fingerprints of reactive nitrogen, so that some of the water masses are clearly discernible by the distribution of nutrients and the isotopic composition of nitrate within the IOSG. Our samples show $NO_3^-/PO_4^{3-}$ ratios

of 14.56 on average (Fig. 8). These $NO_3^-/PO_4^{3-}$ ratios are lower than the global ocean mean of 16:1 (Redfield, 1934, 1963). Furthermore, measurements in the Arabian Sea reveal typical $NO_3^-/PO_4^{3-}$ ratios of 12.81 (Codispoti et al., 2001), even lower than our detected $NO_3^-/PO_4^{3-}$ ratios. Consequently, the average $NO_3^-/PO_4^{3-}$ ratio of 14.56 falls between the global ocean mean of 16:1 (Redfield, 1934, 1963) and the typical ratio in the Arabian Sea of 12.81 (Codispoti et al., 2001). This alone indicates the mixing of water masses of southern and northern Indian Ocean origin.

The deviation from the Redfield stoichiometry (Redfield, 1934, 1963) is quantified by the tracer N*. The analytical error on N* estimate based on the relative error for nitrate and phosphate analysis was below 1.5 % for duplicate sample measurements. The Arabian Sea is characterised by an extensive oxygen deficit zone (ODZ) that induces denitrification in mid-water depths (150−400 m) (Gaye et al., 2013) and leads to an N deficit and therefore to negative N* values (e.g., Bange et al., 2005; Rixen et al., 2005; Gaye et al., 2013). Our data set reveals values of about −1 µmol/kg within the RSPGIW and values lower than −4 µmol/kg within the IDW (Fig. 7d), which coincide with the oxygen minimum (see Sect. 4.1). Consequently, negative N* values are a result of the influx of water masses from the Arabian Sea, which are affected by denitrification. To strengthen this assumption and to compensate the limitations of the N* approach mentioned before, we use stable isotope measurements.

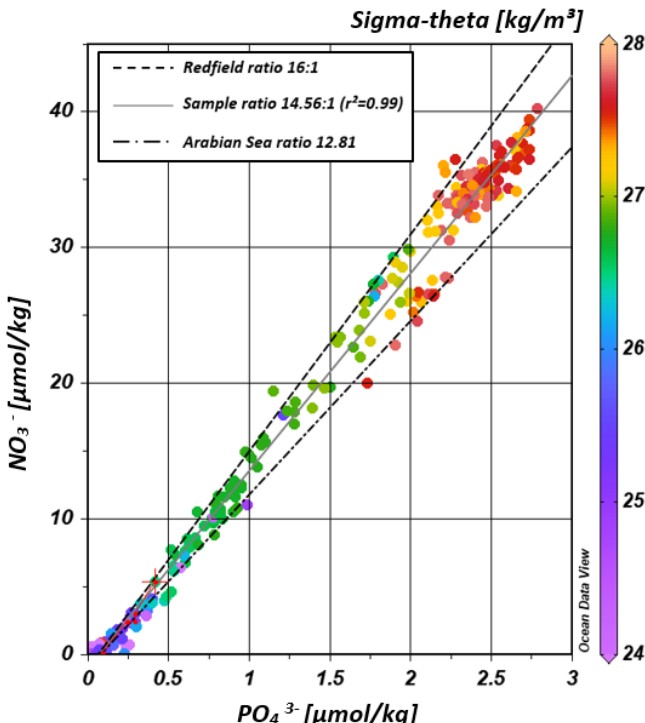

**Figure 8:** Correlation of nitrate versus phosphate concentrations. Regression line of the sample pool (solid, grey line) indicates a ratio of ~14.56 (r²=0.99), intermediate between the Redfield ratio of 16:1 (black, dashes line) and the mean ratio in the Arabian Sea with a slope of ~12.81 (grey, dotted-dashed line) after Codispoti et al. (2001). Colour-coding of dots indicates the potential density sigma theta in kg/m³.

We observe elevated $\delta^{15}$N-NO$_3^-$ and $\delta^{18}$O-NO$_3^-$ values of >7 ‰ and >3 ‰ within the RSPGIW at CTD 01 and 03 (Table 1, Figs. 7e, 7f) accompanied by nitrate concentrations of >30 µmol/kg (Fig. 7a). Denitrification discriminates against the heavier isotope of nitrate ($^{15}$N, $^{18}$O) and raises $\delta^{15}$N-NO$_3^-$ as well as $\delta^{18}$O-NO$_3^-$. In the Arabian Sea average $\delta^{15}$N-NO$_3^-$ and $\delta^{18}$O-NO$_3^-$ values of >20 ‰ and >15 ‰, respectively, are observed in mid-water depth (150−400 m), within the oxygen
minimum zone (Gaye et al., 2013; Martin and Casciotti, 2017). The significant progressive reduction of $\delta^{15}$N-NO$_3^-$ and $\delta^{18}$O-NO$_3^-$ towards the South Indian Ocean is a result of mixing with subtropical thermocline water masses and remineralisation/N-assimilation processes along the flow path.

Within the RSPGIW and the lower IEW, we observe a deviation from the O-to-N isotope effect of $^{18}\varepsilon$:$^{15}\varepsilon$~1 that is typical for consumption processes like denitrification (Granger et al., 2004; Rafter et al., 2013; Sigman et al., 2003; Sigman et al.,
2005). The difference between N and O isotopes ($\delta^{15}$N, $\delta^{18}$O) can be quantified by the tracer $\Delta$(15-18) that indicates values of >4 ‰ within the RSPGIW and the lower IEW (Figure 7g). The RSPGIW injects nitrate that is remineralised from $^{15}$N-enriched organic matter, originated in a region of strong denitrification. During remineralisation of organic matter, the N isotope effect associated with ammonium production and nitrification does not affect the $\delta^{15}$N-NO$_3^-$ but depends on the biomass being remineralised (Rafter et al., 2013). In contrast, the $\delta^{18}$O of newly nitrified nitrate is independent of the
isotopic composition of the organic matter. However, $\delta^{18}$O-NO$_3^-$ depends on the isotope effect during NH$_4^+$ and NO$_2^-$ oxidation, water incorporation ($\delta^{18}$O-H$_2$O of ~0 ‰), and the exchange of oxygen atoms with water that should generate a $\delta^{18}$O of newly produced NO$_3^-$ between -8 and -1 ‰ (Buchwald and Casciotti, 2010; Casciotti et al., 2010). Therefore, the RSPGIW adds nitrate that is enhanced in $\delta^{15}$N compared to the ambient water and has a relative lower $\delta^{18}$O, thus drives the decoupling of N and O isotopes. Furthermore, the source nitrate for N-assimilation in the lower IEW is this regenerated
nitrate and results also in the decoupling of N and O isotopes in this depth range. Consequently, the elevated $\Delta$(15-18) can be explained by a remineralisation/N-assimilation cycle and by the lateral influx of $^{15}$N-enriched nitrate induced by strong denitrification in the oxygen minimum zone of the Arabian Sea. Furthermore, alteration processes within the oxygen minimum zone have the possibility to modify the $\Delta$(15-18). However, it is still unclear how this signal is preserved along the flow path and we suggest that the dominant mechanism that elevates $\Delta$(15-18) values is because of the influx of $^{15}$N-enriched
nitrate due to denitrification.

In the IOSG, we observe elevated $\delta^{15}$N-NO$_3^-$ and $\delta^{18}$O-NO$_3^-$ values of >7 ‰ and >4 ‰ (Table 1, Figs. 7e, 7f) within the SAMW (400−500 m) that is originated in the Subantarctic thermocline of the Southern Ocean. In general, N-assimilation has an isotopic effect of about 5−10 ‰ (Montoya and McCarthy, 1995; Sigman et al., 2005; Waser et al., 1998) and produces biomass that is relatively depleted in $^{15}$N and $^{18}$O in comparison to the nitrate source. Consequently, this drives the elevation
in $\delta^{15}$N and $\delta^{18}$O of the remaining nitrate as uptake proceeds. However, in oligotrophic waters, such as in the IOSG, this isotopic effect is not observable (Montoya et al., 2002) as nitrate is typically drawn down to the limit of detection by complete N-assimilation. Nitrate in surface waters of the Southern Ocean is only partially assimilated due to light limitation and less iron availability (Boyd et al., 2000; DiFiore et al., 2006; DiFiore et al., 2010; Hutchins et al., 2001; Sigman et al.,

1999) and leads to $\delta^{15}$N-NO$_3^-$ values of up to ~13‰ (DiFiore et al., 2006; Sigman et al., 1999, 2000). Seasonal mixing and remineralisation processes result in $\delta^{15}$N-NO$_3^-$ values of 5−9 ‰ within Subantarctic thermocline waters (McCartney, 1977; Sigman et al., 1999). On its flow path towards the north, this isotope trace of incomplete assimilation causes the elevated isotope values within the SAMW that enters the subtropical Indian Ocean thermocline with $\delta^{15}$N-NO$_3^-$ values of >7 ‰
(Table 1, Figs. 4c, 7e, 9a).

Because Subantarctic thermocline waters are the source water of the SAMW and the underlying AAIW in the IOSG, we compare the nitrate isotope properties of Subantarctic thermocline waters with our results. Sigman et al. (1999, 2000) and DiFiore et al. (2006) use the correlation of $\delta^{15}$N-NO$_3^-$ and the fraction of nitrate remaining - ln(NO$_3^-$) - to quantify the isotope fractionation effect during N-assimilation in the Antarctic and Subantarctic region. If N-assimilation occurs with a constant
effect and no new nitrate is added to the surface ocean, then the uptake process can be described in terms of Rayleigh fractionation kinetics (Mariotti et al., 1981). To fulfil the conditions of Rayleigh fractionation, the nitrate samples plot along a straight line in $\delta^{15}$N/ln(NO$_3$) space, where the slope of the line represents the isotope effect of N-uptake or mixing of different nitrate pools. Sigman et al. (1999, 2000) and DiFiore et al. (2006) compare the theoretically Rayleigh utilization trend of $\delta^{15}$N/ln(NO$_3$)=5 ‰ with their measured nitrate utilisation trend within the Subantarctic thermocline, where Sigman
et al., (1999, 2000) determined a slope of $\delta^{15}$N/ln(NO$_3$)~1.3 ‰ (Fig. 9b). Our results reveal a similar, but even shallower slope of $\delta^{15}$N/ln(NO$_3$)~0.93 ‰ (Fig. 9b). It is clearly a mixing signal that causes the moderate slopes of $\delta^{15}$N/ln(NO$_3$) in both the gyre region and in the Subantarctic because biological utilisation of nitrate is unlikely at this depth range. The explanation for the slightly shallower slope in our data set compared to the results in the Subantarctic is vertical mixing with the overlying SSW that has lower $\delta^{15}$N-NO$_3^-$ values (<6 ‰) with nitrate concentrations of <5 µmol/kg. This process does not
fulfil conditions of Rayleigh fractionation because of fundamentally different formation background (Table 1, Fig. 9a; see Sect. 4.2.2). Deep-water nitrate concentrations vary little and $\delta^{15}$N-NO$_3^-$ increases towards shallower water depths and the nitrate signal slightly differ from that of the Southern Ocean due to the influence of the IDW, originated in the Northern Indian Ocean.

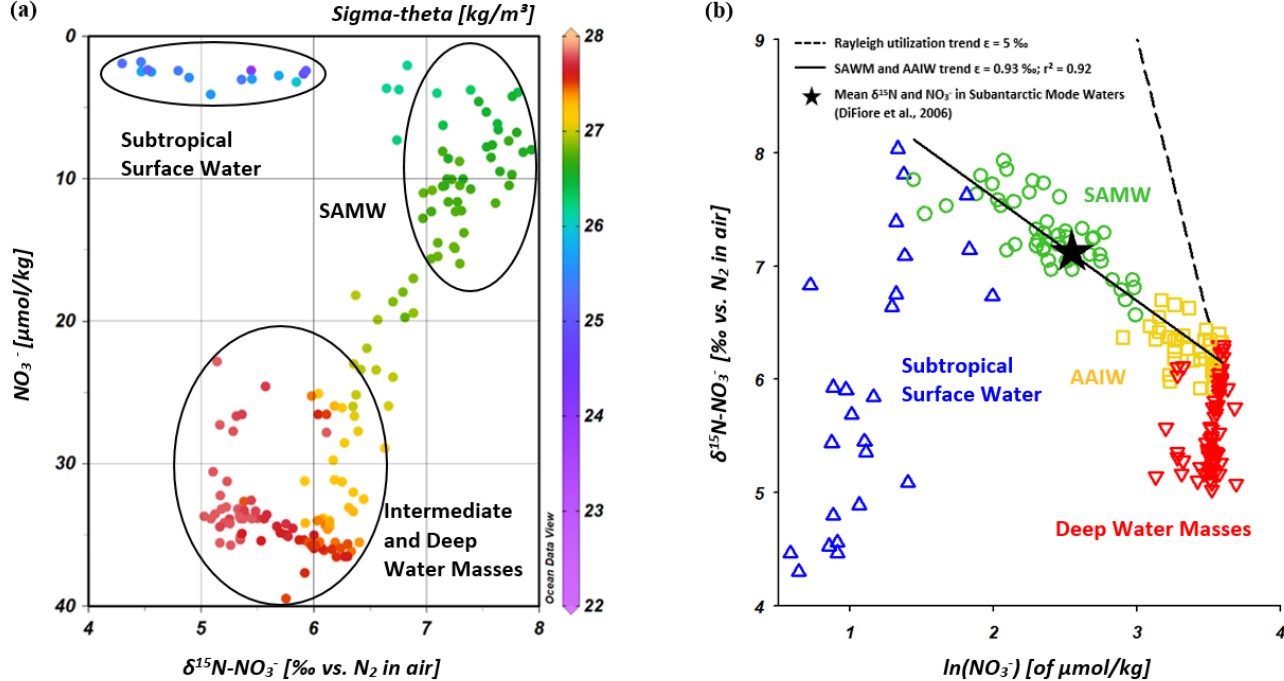

**Figure 9:** Nitrate concentrations versus $\delta^{15}$N-NO$_3^-$ (a) and $\delta^{15}$N-NO$_3^-$ versus ln(NO$_3^-$) (b) for CTD stations within the IOSG (20.36° S–27.78° S). Colour-code of dots in Fig. 9a indicates the potential density sigma-theta (kg/m³). In Fig. 9b, data is grouped for the subtropical surface water (blue), SAMW (green), AAIW (yellow) and deep water masses (red).

Within the isotope maxima of the SAMW (~500 m), the uniform evolution of N and O isotopes of nitrate breaks down and reveals an offset of about 100 m between the N and O isotopic maxima at 23.91° S−20.36° S (Figs. 4c, 4d and see supplement tables S2 and S3). Both, $\delta^{15}$N-NO$_3^-$ and $\delta^{18}$O-NO$_3^-$, are elevated within the SAMW, but $\delta^{15}$N-NO$_3^-$ is less elevated than $\delta^{18}$O-NO$_3^-$. The decoupling of N and O isotopes leads to low $\Delta$(15-18) values of <3 ‰ (Fig. 7g) within the SAMW, that is originated in the partially assimilated Subantarctic region. Isotope fractionation during the initial phase of partial N-assimilation leads to sinking organic matter that is more depleted in [15]N than the source nitrate (Sigman et al., 1999; Rafter et al., 2013). The influx of the SAWM into the subtropical gyre injects [15]N depleted organic matter and its remineralisation adds nitrate that lowers the $\delta^{15}$N of the ambient nitrate and thus leads to low $\Delta$(15-18) in subtropical thermocline waters (Fig. 7g). To conclude, remineralisation of organic matter produced by partial N-assimilation in the Southern Ocean is reflected in lower $\Delta$(15-18) and leads to the deviation of the isotope maxima with an offset of 100 m within the depth range of the SAMW. However, the remineralisation of [15]N-depleted organic matter formed out of newly fixed nitrogen from N$_2$-fixation in surface waters may also influence the decrease of $\Delta$(15-18) (see the following Sect. 4.2.2).

#### 4.2.2 Evidence for N₂-fixation in the IOSG

The mixing of source water signals from the lateral influx of the neighbouring northern Indian Ocean and Southern Ocean significantly affects the nutrient distribution and isotopic composition of nitrate in the gyre region. However, N* and $\delta^{15}$N-NO$_3^-$ suggest that N₂-fixation introduces new nitrate into the surface waters of the IOSG. The increase in N* up to 2 µmol/kg

at 60−200 m indicates a positive deviation of the NO$_3^-$/PO$_4^{3-}$ ratio from the Redfield stoichiometry (Gruber and Sarmiento, 1997; Redfield, 1934, 1963) and is an evidence for the input of newly fixed N into the surface water. Studies in the South-West Pacific Ocean also indicate positive N* anomalies of ~2 µmol/kg in the productive layer reflecting diazotrophic N₂-fixation (Fumenia et al., 2018). Although N₂-fixation is the first order driver of positive N* (Bourbonnais et al., 2009; Monteiro and Follows, 2006), other processes, for instance atmospheric deposition or the preferential remineralisation of N

over P, may also be responsible for excess N and result in an overestimation of N*-based N₂-fixation (Bourbonnais et al., 2009; Monteiro and Follows, 2006). However, the South Indian Ocean is less affected by the influx of nutrient enriched mineral aerosols from atmospheric deposition (Duce at al., 2008; Duce and Tindale, 1991) and we neglect this factor in our further discussion. Furthermore, we use $\delta^{15}$N and $\delta^{18}$O of nitrate, as well as $\Delta$(15-18), as additional indicators of N sources to overcome the weakness associated with the N* approach.

Diazotrophic N₂-fixation produces organic matter that has a low $\delta^{15}$N-NO$_3^-$ relative to average oceanic combined nitrogen (Carpenter et al., 1997; Minagawa and Wada, 1986; Montoya et al., 2002; Wada and Hattori, 1976). Within the upper 200 m of the IOSG (20.36° S−23.91° S), the $\delta^{15}$N of nitrate varies between 4.3 ‰ and 5.9 ‰ with a mean of ~5.0 ‰ (Figs. 7e, 9a, 9b). These values are higher compared to other regions of intense N₂-fixation, such as in the subtropical NE Atlantic, where values of 2–5 ‰ occur in surface waters (Bourbonnais et al., 2009). At first sight, the high values of surface waters in the

IOSG do not speak for significant N₂-fixation in surface waters. However, considering that SAMW is the source of nitrate with elevated $\delta^{15}$N-NO$_3^-$ values of 7.4 ‰ on average (20.36° S−23.91° S) and highest values of 7.9 ‰ (Figs. 9a, 9b, see also supplement table S2), the $\delta^{15}$N-NO$_3^-$ in surface waters (4.3–5.9 ‰) is lowered by ~2.4 ‰. This is similar to the decrease of ~3 ‰ in surface waters of the north Atlantic, where N₂-fixation is verified (Bourbonnais et al., 2009; Knapp et al., 2008). Therefore, N₂-fixation is most likely the main driver in the upward decrease of $\delta^{15}$N-NO$_3^-$ in surface waters and adds

isotopically light nitrogen from the atmosphere into the gyre region. To prove this, we take the nitrate $\delta^{18}$O into account, which exhibits values of <3 ‰ and shows a minor decrease compared to $\delta^{15}$N. The resulting decoupling of N and O isotopes of nitrate leads to smaller differences between $\delta^{15}$N and $\delta^{18}$O and reveals a $\Delta$(15-18) of <3 ‰ (Fig. 7g). To conclude, positive N* values, distinct upward decrease of $\delta^{15}$N-NO$_3^-$, reduced $\Delta$(15-18) and the distance from any external nitrate sources are unambiguous evidences of diazotrophic activity within the surface layer.

To estimate the supply of newly fixed N to the nitrate pool within the IOSG, we can calculate the fraction of nitrate coming from atmospheric N₂-fixation and the fraction that is added from the underlying source water by using the observed $\delta^{15}$N-NO$_3^-$ within the upper 200 m using the following equation modified after Bourbonnais et al. (2009; 2013):

$$\delta^{15}N_{surface} = \left(\delta^{15}N_{fix} \times a\right) + \left(\delta^{15}N_{source} \times b\right) \tag{1}$$

where "$\delta^{15}N_{surface}$" is the detected $\delta^{15}N\text{-}NO_3^-$, which shows an average of ~5.0 ‰ in the upper 200 m (range 4.3–5.9 ‰) at latitude 20.36° S−23.91° S. The "$\delta^{15}N_{fix}$" is the isotope value of atmospheric $N_2$, assumed to be about 0 ‰ and the factor "a" is the percentage of nitrate coming from atmospheric $N_2$-fixation. The $\delta^{15}N\text{-}NO_3^-$ of the source water, which is the SAMW, with values about 7.4 ‰, is represented by "$\delta^{15}N_{source}$" and "b" is the corresponding percentage.

In this equation we neglect the point of isotope fractionation via N-assimilation due to the fact, that in oligotrophic environments a complete N-assimilation takes place (Montoya et al., 2002). Thus, no net expression of the isotope fractionation occurs. Consequently, the produced organic matter that is again being remineralised also has a similar isotope signal as the assimilated nitrate. This assimilated nitrate is the mixing product of newly fixed nitrogen from the atmosphere and the input from the underlying SAMW, which is expressed by equation 1. We resolve the equation to a (=1−b) and b (=$\delta^{15}N_{surface}$ /$\delta^{15}N_{source}$) and hence we calculate that about 32 % of the assimilated nitrate is provided from newly fixed nitrogen by $N_2$-fixation.

To prove this first approach, indicating that about 32 % of the assimilated nitrate results from atmospheric N input by $N_2$-fixation, we can calculate the $NO_3^-/PO_4^{3-}$ ratio resulting from nutrient assimilation without any external N input by $N_2$-fixation and compare this with our measured $NO_3^-/PO_4^{3-}$ ratio within surface waters. To provide an estimate of the excess nitrogen in surface waters supplied by the remineralisation of cyanobacterial biomass, we use the following equations:

$$NO_3^-/PO_4^{3-}{}_{cal} = NO_3^-{}_{sample}/\left(\frac{NO_3^-{}_{in}}{NO_3^-/PO_4^{3-}{}_{in}} - \frac{NO_3^-{}_{ass}}{NO_3^-/PO_4^{3-}{}_{ass}}\right) \tag{2}$$

with "$NO_3^-{}_{in}$" as initial nitrate concentration of the source water (SAMW within the IOSG), "$NO_3^-{}_{ass}$" denoting the assimilated nitrate ($NO_3^-{}_{in}-NO_3^-{}_{sample}$) and "$NO_3^-{}_{sample}$" being the sample concentrations. The initial nitrate to phosphate pool $NO_3^-/PO_4^{3-}{}_{in}$ is defined as mean ratio of the source water. For the region of the IOSG, we presume that the mean ratio within the SAMW is 13.25. For the general N-assimilation in the euphotic zone we assume Redfield conditions of $NO_3^-/PO_4^{3-}{}_{ass}$ = 16. To calculate the residual nitrate, we multiply the calculated nitrate to phosphate ratio ($NO_3^-/PO_4^{3-}{}_{cal}$) with the measured phosphate concentrations:

$$NO_3^-{}_{cal} = NO_3^-/PO_4^{3-}{}_{cal} * PO_4^{3-}{}_{sample} \tag{3}$$

The difference of $NO_3^-{}_{sample}$ and $NO_3^-{}_{cal}$ represents the portion of the nitrate supplied by nitrification out of newly fixed N. At latitude 20.36° S−23.91° S, our samples indicate elevated $NO_3^-/PO_4^{3-}$ ratios and a resulting positive deviation from the calculated line of N-assimilation at nitrate concentrations of <10 µmol/kg (Fig. 10a), that indicates an external input of N into the surface waters of the IOSG. We presume that $N_2$-fixation leads to the local elevation in $NO_3^-/PO_4^{3-}$ ratios due to the input of new N and coincide with the decrease of $\delta^{15}N\text{-}NO_3^-$ and the decoupling of N and O isotopes, leading to low $\Delta$(15-18). The quantity of newly fixed nitrate ($NO_3^-{}_{new}$) is given by the formula:

$$NO^-_{3\,new}\ [in\ \%] = \frac{(NO^-_{3\,sample} - NO^-_{3\,cal})}{NO^-_{3\,sample}} * 100\ , \tag{4}$$

which is presented in Fig. 10b, indicating a distinct upward increase in the upper 200 m at 20.36° S−23.91° S with an average portion of fixed nitrate of about 34 %. Consequently, our first approach that suggests that 32 % of the nitrate measured in the upper 200 m is derived from newly fixed nitrogen hence agrees quite well with the 34 % calculated by using

$NO_3^-/PO_4^{3-}$ ratios. Bourbonnais et al. (2009) stated that $N_2$-fixation accounts for ~40 % of newly supplied nitrate in the subtropical North Atlantic. This is slightly higher than our assumption for the subtropical South Indian Ocean. However, in the subtropical North Atlantic higher N* values (3.5 µmol/kg), a slightly stronger upward decrease of $\delta^{15}N$-$NO_3^-$ and a stronger decoupling of N and O isotopes are observed in surface waters, suggesting higher fixation rates.

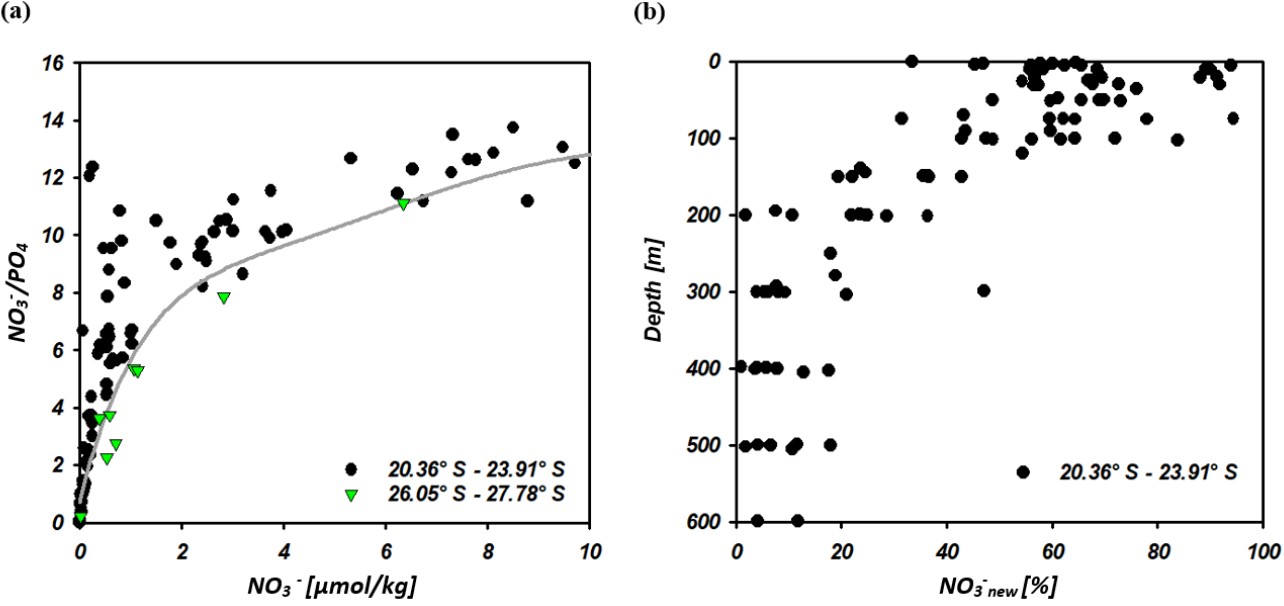

**Figure 10:** $NO_3^-/PO_4^{3-}$ ratio versus nitrate concentrations of seawater samples at 20.36° S−23.91° S and 26.05-27.78° S (a). The grey solid line indicates the calculated N-assimilation ($NO_3^-{}_{cal}$ vs. $NO_3^-/PO_4^{3-}{}_{cal}$) with a preformed $NO_3^-$:$PO_4^{3-}{}_{in}$ ratio of 13.25 for the region of the IOSG and 14.25 for the southern equatorial Indian Ocean and progressive nutrient assimilation with a Redfield ratio of 16 ($NO_3^-/PO_4^{3-}{}_{ass}$). In Fig. 10b we present the portion of nitrate formed out of newly fixed N ($NO_3^-{}_{new}$) versus depth at latitudes 20.36° S−23.91° S.

Further south at 26.05−27.78° S, samples plot close to the line of N-assimilation and no significant input of fixed nitrate is

indicated (Fig. 10a). This agrees with the $\delta^{15}N$-$NO_3^-$ values in surface waters, which demonstrate an abrupt increase at about 26° S to similar values as in the underlying SAMW (>7 ‰), while $\delta^{18}O$-$NO_3^-$ shows still low values of <3 ‰. Consequently, N and O isotopes reveal a counteracting behaviour that differs from the region at 20.36° S−23.91° S, resulting in high Δ(15-18) of >4.5 ‰ (Fig. 7g). This is a strong indication for the absence of $N_2$-fixation in this region but also leads to the assumption that $\delta^{18}O$ remains low due to ongoing nitrate production by nitrification. This sudden termination of $N_2$-fixation

may be due to a temperature-limiting factor, mentioned by Capone et al. (1997) and Breitbarth et al. (2007). They argue that $N_2$-fixation by *Trichodesmium*, which is the dominant nitrogen-fixing cyanobacteria in subtropical oligotrophic waters (Berman-Frank et al., 2001; Pearl et al., 1994), decreases dramatically at seawater temperatures below 22°C (Berman-Frank et al., 2001). Modelling $N_2$-fixation with this assumption thus resulted in very low fixation rates south of about 25° S in the

Indian Ocean (Paulsen et al., 2017). However, $N_2$-fixation by other diazotrophs (e.g., unicellular diazotrophic cyanobacteria) has been shown to occur at higher latitudes than *Trichodesmium* (Moisander et al., 2010). Another reason for the decline of $N_2$-fixation south of 26° S may the limited availability of iron and other micronutrients. Atmospheric iron deposition is low in the southern hemisphere oceans and iron availability gradually decreases towards high southern latitudes (Boyd et al., 2000; Duce and Tindale, 1991; Duce et al., 2008). Reduced iron availability is suggested to limit growth of nitrogen-fixing

organisms in regions of already limited iron availability (Sanudo-Wilhelmy et al., 2001). Berman-Frank et al. (2001) calculated the potential of nitrogen fixation by *Trichodesmium* and suggested that in 75% of the global ocean, iron availability limits nitrogen fixation. However, until now no concrete studies on iron and other micronutrient availability and $N_2$-fixation have been conducted within the South Indian Ocean.

**Conclusion**

The South Indian Ocean gyre is the only oligotrophic gyre in the Indian Ocean due to the land-locked nature of the northern Indian Ocean. Compared to the Atlantic and Pacific Ocean gyres the IOSG is less explored and is poorly understood in terms of nutrient distribution and isotopic composition of nitrate.

This work compiles the general distribution of water masses from 30° S, within the IOSG, across the South Equatorial Current (SEC), and towards the southern equatorial Indian Ocean. We established the first water mass distribution model in

this ocean region that provides a basis for the identification of nutrient sources and the isotopic signatures of nitrate. Water masses in our study area are diverse and originate in two fundamentally different ocean regimes: the Southern Ocean (SAMW and AAIW) and the northern Indian Ocean (RSPGIW and IDW). These different water masses have a major influence on the nutrient distribution and stable isotope composition of nitrate in the IOSG.

Our nutrient and isotopic data, which are one of the first reported for the subtropical South Indian Ocean, demonstrate the

lateral influx from the Arabian Sea, characterised by strong denitrification in mid-water depths that leads to an N deficit in intermediate and deep waters accompanied by elevated isotope ratios of nitrate within the RSPGIW. The lateral influx from the Southern Ocean is via the oxygen saturated SAMW, with characteristically elevated isotope ratios of nitrate due to partial N-assimilation in high southern latitudes. Additionally, our data mirror an external input of N by $N_2$-fixation that is indicated by positive N* and low Δ(15-18) values in surface waters. In the upper 200 m in the region of 20.36° S−23.91° S, we

calculate that approximately 32–34 % of the nitrate consumed by N-assimilation is provided from newly fixed nitrogen.

The IOSG has been sparsely investigated and is an area representing those oceanic oligotrophic regions that are likely to adjust to continued warming by deepening stratification, reduced upward nutrient supply across the thermocline, and decreasing biological production. Whether this will be offset by enhanced $N_2$-fixation in warming layers remains as an open question that needs dedicated follow-up studies, i.e., in terms of experimental approaches, time series observation, remote

sensing, and biogeochemical modelling.

*Data availability.* All data of cruises SO 259 and MSM 59/2 are available at the PANGAEA data publisher for Earth & Environmental Science. The data can be found under Harms, Natalie; Lahajnar, Niko; Gaye, Birgit; Rixen, Tim; Dähnke, Kirstin; Ankele, Markus; Schwarz-Schampera, Ulrich; Emeis, Kay-Christian (2019): Physical oceanography, nutrients, and nitrogen and oxygen isotopic composition of nitrate measured on water bottle samples during Maria S. Merian cruise

MSM59/2. PANGAEA, https://doi.pangaea.de/10.1594/PANGAEA.897503 and Harms, Natalie; Lahajnar, Niko; Gaye, Birgit; Rixen, Tim; Dähnke, Kirstin; Ankele, Markus; Schwarz-Schampera, Ulrich; Emeis, Kay-Christian (2019): Physical oceanography, nutrients, and nitrogen and oxygen isotopic composition of nitrate measured on water bottle samples during SONNE cruise SO259. PANGAEA, https://doi.pangaea.de/10.1594/PANGAEA.897504.

*Author contribution.* N. Lahajnar and N. C. Harms collected the samples on board. U. Schwarz-Schampera conceived the

INDEX program in the IOSG and led the cruises. N. Lahajnar, B. Gaye, T. Rixen and K.-C. Emeis designed the nutrient and nitrogen cycle study. N. C. Harms, N. Lahajnar, K. Dähnke and M. Ankele participated in the sample analyses. N. C. Harms, N. Lahajnar and B. Gaye analysed the data. N. C. Harms wrote the first draft of the manuscript. All authors contributed substantially to the final paper

*Competing interests.* The authors declare that they have no conflict of interest.

*Acknowledgments.* Cruises and sampling were conducted within the framework of the INDEX program of the Federal Institute for Geosciences and Natural Resources (BGR). The INDEX program explores polymetallic sulfides on the ocean floor, based on a fifteen-year contract of BGR with the International Seabed Authority. BGR requests acknowledgment in any future use of the data and results in this publication. We thank the crew of the German research vessels *Maria S. Merian* and *Sonne* for their outstanding support of our work on board. Furthermore, we thank our colleagues from the Helmholtz

Institute Geesthacht, especially Tina Sanders, for supporting our analyses of nutrients and stable isotopes of nitrate.

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
