# Peer review of "Nutrient distribution and nitrogen and oxygen isotopic composition of nitrate in water masses of the subtropical South Indian Ocean"

_Biogeosciences, 2018_

## Referee Comment (RC1) · Bourbonnais (Referee) · 30 Jan 2019

**General comments**

Harms *et al.* report concentrations of water-column nutrients and stable isotope composition of nitrate for the subtropical South Indian Ocean. They discuss their results in relation to the different water masses and estimate that one third of the nitrate in the upper ocean is supplied by $N_2$ fixation in this region. While, the study is interesting as few isotopic data exist for the subtropical Indian Ocean, it is difficult to appreciate the new findings from their discussion. Second, they estimate the contribution from $N_2$ fixation using N*, and assuming a Redfield ratio of 16, which might not be valid in a region where $N_2$ fixers are abundant. They do not discuss other N sources (e.g., atmospheric depositions). More importantly, their interpretation would benefit from better exploiting the information derived from the dual isotopic composition of nitrate, perhaps using a simple isotope box model, as in Knapp *et al.* (2008) and Bourbonnais *et al.* (2009).

**Specific comments**

**Abstract**

Generally, the abstract should better indicate what are the new findings.

Page 1, line 20: N* < 1 µM is not a strong N deficit relatively to other regions of the ocean where N deficit is close to 40 µM (see Bourbonnais *et al*., 2015). What is the analytical error on their N* estimate? Also, indicate depth of the minimum N*.

Page 1, lines 23-24: Indicate how the contribution from $N_2$ fixation was estimated (i.e., using N:P and Redfield ratio assumptions).

**Introduction**

Page 2, line 12: The transition is awkward. Rewrite.

Page 2, lines 20-22: One important caveat is that N* cannot be used to derive rates of $N_2$ fixation in region where denitrification co-occurs, as the N* signatures associated with denitrification and $N_2$ fixation are overprinting each other's. One advantage of measuring the dual isotopic composition of nitrate is that it allows disentangling these different overprinting processes, because, as stated later in the manuscript, $N_2$ fixation is associated with negative N to O nitrate isotope anomalies. On the other hand, denitrification is not expected to produce such N to O nitrate isotope anomalies because N and O are equally fractionated during this process. This point should be better emphasized in the introduction (and better exploited in their discussion).

Page 2, lines 28-29: change for: "lighter isotopes are preferentially assimilated, leaving the substrate enriched in $^{15}N$ and $^{18}O$."

Page 3, line 1: Add references here, e.g., Knapp *et al*., 2008 and Bourbonnais *et al*., 2009.

Page 3, line 4: Which depth range corresponds to $\delta^{15}N_{deep}$ and $\delta^{18}O_{deep}$?

Page 3, lines 14-16: Be more specific about the new findings from this study. Which specific gaps were filled comparatively to previous studies?

**Materials and Methods**

Page 6, line 13: Why using a single point correction only?

Page 6, line 15: What was blank size?

**Results:**

Page 6, section 3.1: It would be helpful to show T-S diagrams at this point rather than later in the discussion.

Page 7, lines 6-16: Figure 5 (panels a, b, c, d) should be presented in this section and table 2 moved to the supplementary materials.

Page 8, lines 2-13: Figure 5 (panels e, and f) should be presented in this section and table 3 moved to the supplementary materials.

**Discussion**

Page 9, line 6-7: What is new in their water mass distribution model (Figure 4)?

Page 9, lines 25-26: Change for "… because of respiration and the absence of effective ventilation…"

Pages 9-12: It would be useful to include the nitrate isotopic composition (end-members) for the different water masses, either in figure 4, or in a table.

Page 12, lines 20-21: The $NO_3^-/PO_4^{3-}$ should however increase if $N_2$ fixation is significant.

Page 13, lines 10-11: How does the mean $NO_3^-/PO_4^{3-}$ ratio changes along the latitudinal transect? What are the implications for $N_2$ fixation?

Page 14, line 1: Bourbonnais *et al.* (2009) is incorrectly referenced here.

Page 15, lines 5-6 : Add references to support this statement.

Page 16, lines 17-18: Why nitrate utilization is unlikely? It is too deep?

Page 18, line 16 : Bourbonnais *et al.* (2009) is once again incorrectly referenced in this context.

Page 18, lines 21-24: Bourbonnais *et al.* (2009) report a range of 2 to 5‰ for the $\delta^{15}N$ of nitrate in surface waters of the subtropical northeast Atlantic Ocean. Using a simple isotopic mass balance, they estimated that $N_2$ fixation could account for up to 40% of the export production in this region.

Page 18, lines 29-31: It is peculiar to note that the $\Delta(15,18)$ anomalies observed in this studies are at least half of the anomalies observed in the subtropical northeast Atlantic Ocean by Bourbonnais *et al.* (2009) ($\Delta(15,18)$ of -7 to 0‰). Why would that be if the estimated contribution from $N_2$ fixation is supposedly in the same range (accounting for 30-40% of new supplied nitrate) for these two regions? The N* observed by Bourbonnais et al. (2009) was also up to ~3.5 µmol/kg.

Page 19, lines 8-9: In equation (6), the nitrate to phosphate ratio ($NO_3^-/PO_4^{3-}$) is divided by the measured phosphate concentrations, not multiplied.

Page 19, lines 1-23: This approach requires many assumptions. One likely invalid assumption is assuming a Redfield ratio of 16. The Redfield ratio is variable in marine microalgae (see Geider *et al.*, 2002). $N_2$ fixers also have higher N:P ratios (e.g., Letelier *et al.*, 1998). Finally, this approach does not take into account inputs from atmospheric depositions.

Page 19, line 21: This is confusing, as $\delta^{15}N\text{-}NO_3^-{}_{fix}$ (i.e. supplied from $N_2$ fixation) should be about 0‰. I suggest removing the "fix" subscript.

Page 19, lines 20-23: Overall, the dual nitrate isotopic data could be better exploited in their discussion and used in an isotopic box model to derive an independent assessment of the contribution from $N_2$ fixation (see examples from Knapp *et al.*, 2008 and Bourbonnais *et al.*, 2009).

Page 20, lines 9-10: Bourbonnais *et al.* (2009) did not observe significant positive $\Delta(15,18)$ anomalies in the subtropical northeast Atlantic Ocean. Which make me wonder what is the propagated (analytical) error associated with their $\Delta(15,18)$ measurements. In other words, is their calculated positive $\Delta(15,18)$ significantly different from 0?

Page 20, line 16: $N_2$ fixation have been shown to occur at lower temperatures in temperate regions (see Moisander *et al.*, 2010).

**Tables**

**Table 1** is not necessary since the information is already presented in Figure 1. I recommend moving it to the supplementary materials.

**Table 2** should be moved to the supplementary materials as this information is already in Figures 3 and 5.

**Table 3** should be moved to the supplementary materials as this information is already in Figure 5.

**Figures**

**Figure 1:** It is difficult to see the shaded arrow representing the South equatorial current.

**Figure 6:** What is the $r^2$ and error on the slope?

**Figure 7b:** Which processes cause the positive $\Delta(15,18)$?

**Technical comments**

Page 1, lines 30-31: this sentence is repetitive. Replace by something like: "The South Indian Ocean is dominated by a subtropical anticyclonic gyre (refs), the Indian Ocean subtropical gyre" (IOSG), one of the major subtropical gyres in the world's ocean. The IOSG has been, thus far, sparsely investigated."

Page 2: line 2: Use the IOSG acronym defined earlier.

Page 3, line 10: Remove "Therefore" at the beginning of sentence.

Page 12, line 7: change for " nutrient distribution and N cycle processes"

**Additional references (not already in cited literature):**

Bourbonnais, A., Altabet, M. A., Charoenpong, C. N., Larkum, J., Hu, H., Bange, H. W., & Stramma, L. (2015). N-loss isotope effects in the Peru oxygen minimum zone studied using a mesoscale eddy as a natural tracer experiment. *Global Biogeochemical Cycles*, *29*(6), 793-811.

Geider, R. J., & La Roche, J. (2002). Redfield revisited: variability of C [ratio] N [ratio] P in marine microalgae and its biochemical basis. *European Journal of Phycology*, *37*(1), 1-17.

Knapp, A. N., DiFiore, P. J., Deutsch, C., Sigman, D. M., & Lipschultz, F. (2008). Nitrate isotopic composition between Bermuda and Puerto Rico: Implications for N2 fixation in the Atlantic Ocean. *Global Biogeochemical Cycles*, *22*(3).

Letelier, R. M., & Karl, D. M. (1998). Trichodesmium spp. physiology and nutrient fluxes in the North Pacific subtropical gyre. *Aquatic Microbial Ecology*, *15*(3), 265-276.

Moisander, P. H., Beinart, R. A., Hewson, I., White, A. E., Johnson, K. S., Carlson, C. A., ... & Zehr, J. P. (2010). Unicellular cyanobacterial distributions broaden the oceanic N2 fixation domain. *Science*, *327*(5972), 1512-1514.

---

## Referee Comment (RC2) · Anonymous Referee #2 · 11 Feb 2019

This is an interesting paper, with a lot of valuable information from a region that has not had much attention with respect to nitrate isotopes. The connection to the water mass structure was well done. There are some points that require clarification, especially the calculation and discussion of nitrogen fixation inputs. These comments and others are detailed below.

Abstract

Page 1, line 20: I would remove 'strong' here, as N* of -1 $\mu$M would not generally be considered "a strong N deficit".

Page 1, lines 21-23: Please clarify what you are referring to here using "preformed versus regenerated". The preceding sentence referred to nitrate isotope signals coming from SAMW and from denitrification in the Arabian Sea. Where is the 'regenerated' signal that you are referring to?

Page 1, lines 23-25: If there is significant N2 fixation, I would not expect low nitrate to phosphate ratios. Revisit the N2 fixation discussion below.

Introduction

Page 2, line 21: I think a reference to Gruber and Sarmiento, 1997 would be appropriate here.

Page 2, lines 29-30: The isotopic fractionation factor, $\varepsilon$, relates the instantaneous product, not the accumulated product, to the substrate. Though neither is explicitly stated, I think the implication is that this always holds true. This should be clarified.

Materials and Methods

Page 6, line 14: How is the 'single point correction' for $\delta$15N applied? Is this simply a standard subtraction?

Results

Page 6, line 29: What water mass does the 34.6 PSU feature represent?

Figure 2: It might be more helpful to include contours for the potential density surfaces, rather than contouring the same properties represented on the color bar.

Pages 7 and 8: I don't understand the choices behind what is shown in Tables 2 and 3. Why are these specific density/depth intervals selected, and why look at different density levels in the different latitude zones? Why are only one nitrate and phosphate concentration (Table 2) or nitrate $\delta$15N and $\delta$18O value (Table 3) given for each line? How many measurements are included in these values? Shouldn't there be a range or uncertainty given for these if they derive from a range in latitude?

Throughout this presentation of results in sections 3.2 and 3.3, I found referring to Figure 5 more useful than consulting Tables 2 and 3. I would suggest moving Figure 5 earlier in the paper, and removing Tables 2 and 3, or perhaps moving them to the supplement, unless their relevance can be better explained.

Page 7, line 6: I would delete 'strongly'. When working in oligotrophic areas, I'm not sure 5.9 $\mu$M nitrate qualifies as "strongly depleted". Otherwise, you could perhaps cite the concentration of nitrate in the surface waters, rather than at 310 m.

Discussion

Page 10, line 27: Please clarify "decrease of the oxygen minimum". Do you mean that the oxygen concentration is increasing? If so, please rephrase.

Figure 3: I didn't find this figure necessary, and suggest that it be moved to the supplement.

Figure 4: I think this figure is extremely helpful for thinking about the water mass structure of the region! My only question is what determines where the lines dividing water masses are drawn? Are these specific sigma theta surfaces? Please clarify.

Figure 6: The figure legend states that the color bar indicates potential density, but what is actually used is depth. Perhaps sigma theta would, in fact, be better.

Page 15, line 16: Doesn't iron availability also play a role in incomplete nitrate assimilation in the Southern Ocean?

Page 17, line 3: Please clarify "lower water depths". Do you mean shallower or deeper?

Figure 8: The yellow star representing the mean nitrate $\delta$15N does not stand out. I would suggest making this symbol a different color or shape. Also, please provide the slope of the solid line in the figure legend.

Page 17, line 11: Typo, should be 'SAMW' rather than "SAWM".

Page 17, line 14: Please give Sigman et al., 2005 reference to Δ(15,18). Rafter et al (2013) is also a good reference, but uses Δ(15-18) instead.

Page 18, line 16: I would include a reference here to Gruber and Sarmiento 1997 for their seminal work in this area.

Page 18, line 24: One could also reference work in the Atlantic from Knapp et al., 2008, and a variety of work from the Pacific.

Page 19, lines 7-8: What are the implications of assuming Redfield stoichiometry here?

Page 19, line 10: This equation appears incomplete, if not incorrect. From the text, I would not expect PO43-sample to appear in the denominator.

Page 19, line 12: What is the N:P ratio assumed for newly fixed N? This seems important to the calculations performed here.

Page 19, lines 20-23: A newly fixed $\delta$15N of 4.8‰ is not within the range of expected values for N2 fixation. This seems problematic, and requires reevaluation and justification of the approach used to arrive at this value. In my mind, a value of +4.8‰ argues against this N deriving from N2 fixation. What other explanations have the authors considered?

Figure 9: It is difficult to distinguish the symbols used to represent the two geographic areas in panel a. What calculation is used to derive the gray line in panel a?

Page 20, line 16: Is low temperature the only other possible explanation? Increasing numbers of reports are finding N2 fixation at low temperature, thus the temperature limits seem to be a less convincing argument. What other contributing factors could be here?

Page 21, lines 10-11: Can you make any connection here to the results of Martin and Casciotti, 2017 from the Arabian Sea?

---

## Author Response (AR1)

**Reply to the Review of the Referee #1 (Bourbonnais)**

(**RC**: Referee Comment; **AR**: Author's Responds)

First of all, thank you very much for thoroughly reviewing our manuscript and for the helpful comments and suggestions. We try to include as many comments and suggestions as possible which help us to improve our manuscript.

*Abstract*

**Page 1, line 20**

> **RC:** $N^*$ <-1 µM is not a strong N deficit relatively to other regions of the ocean where N deficit is close to 40 µM (see Bourbonnais et al., 2015). What is the analytical error on their $N^*$ estimate? Also, indicate depth of the minimum $N^*$.

> **AR:** I agree with you that in comparison with other regions an $N^*$ of <-1 µM is not really a strong N deficit, but with lowest values of -4 µM within the IDW (Page 14, lines 4-5) these values are significant in our study area. The analytical error on our $N^*$ estimate based on the relative error of nitrate and phosphate analyses was below 1.5 % for duplicate sample measurements (Page 5, lines 12-13). The $N^*$ Minimum is located within the RSPGIW and the IDW at a core depth of ~1500 m and ranged from ~1000 m until ~1600 m (Page 5, lines 12-13). This information will be added to the abstract.

**Page 1, lines 23-24**

> **RC:** Indicate how the contribution from $N_2$-fixation was estimated (i.e., using N/P and Redfield ratio assumptions).

> **AR:** We are using a simple calculation for a first estimate of the input of new nitrate into the surface layer by $N_2$-fixation by using the deviation of the N/P-Redfield-ratio. We will clarify this in the revised version.

*Introduction*

**Page 2, line 12**

> **RC:** The transition is awkward. Rewrite.

> **AR:** We modified the sentence to: "To study the marine nitrogen cycle, we use nitrate and phosphate concentrations as well as the isotopic signature of nitrate (Deutsch et al., 2001; Deutsch et al., 2007; Gruber and Sarmiento, 1997; Lehmann et al., 2005; Sigman et al., 2005)."

**Page 2, lines 20-22**

> **RC:** One important caveat is that $N^*$ cannot be used to derive rates of $N_2$-fixation in region where denitrification co-occurs, as the $N^*$ signatures associated with denitrification and $N_2$-fixation are overprinting each other's. One advantage of measuring the dual isotopic composition of nitrate is that it allows disentangling these different overprinting processes, because, as stated later in the

manuscript, N$_2$-fixation is associated with negative N to O nitrate isotope anomalies. On the other hand, denitrification is not expected to produce such N to O nitrate isotope anomalies because N and O are equally fractionated during this process. This point should be better emphasized in the introduction (and better exploited in their discussion).

**AR:** You are right that in regions where denitrification and N$_2$-fixation simultaneously occur N* cannot be used alone. However, in our study area no denitrification takes place and we just see a signal in intermediate and deep waters coming from the Arabian Sea, where denitrification take place. We use the positive surface N* signatures as a first evidence for N$_2$-fixation and confirm these signatures with the distinct upward decrease of N-isotope values compared to strongly elevated $\delta^{15}$N values in subsurface waters (~500 m, elevation of ~2-3.5 ‰). I agree that dual isotope measurements of nitrate will help to improve the weakness associated with the N* approach and we will rewrite and add a section on dual isotopes.

**Page 2, lines 28-29**

**RC:** Change for: "lighter isotopes are preferentially assimilated, leaving the substrate enriched in $^{15}$N and $^{18}$O."

**AR:** This sentence will be rewritten as you noted.

**Page 3, line 1**

**RC:** Add references here, e.g., Knapp et al., 2008 and Bourbonnais et al., 2009.

**AR:** We will add these references.

**Page 3, line 4**

**RC:** Which depth range corresponds to $\delta^{15}$Ndeep and $\delta^{18}$Odeep?

**AR:** For $\delta^{15}$Ndeep and $\delta^{18}$Odeep we use the mean of $\delta^{15}$N and $\delta^{18}$O within the water depth below 2000 m. When we will still use the tracer $\Delta(15,18)$ (see explanation below) we will add this information in the introduction part and in the discussion section on Page 18, line 1.

**Page 3, lines 14-16**

**RC:** Be more specific about the new findings from this study. Which specific gaps were filled comparatively to previous studies?

**AR:** We will explain more precisely that our findings filled the gaps between the mentioned studies relating to nutrient distribution, nitrate isotope measurements and water mass analyses. First, in this region, we linked the different water masses of different origin with their isotopic signature. We will clarify our new findings in the revised version.

*Materials and Methods*

**Page 6, line 13**

**RC:** Why using a single point correction only?

**AC:** We will correct the method section, because we indeed do not use a single point correction but rather a two-point correction referred to IAEA-N3 ($\delta^{15}$N-NO$_3^-$ = +4.7 ‰ and $\delta^{18}$O-NO$_3^-$ = +25.6 ‰) and USGS-34 ($\delta^{15}$N-NO$_3^-$ = −1.8 ‰ and $\delta^{18}$O-NO$_3^-$ = −27.9 ‰) for $\delta^{15}$N-NO$_3^-$ and $\delta^{18}$O-NO$_3^-$.

**Page 6, line 15**

**RC:** What was blank size?

**AR:** The standard deviation for IAEA-N3 was generally better than 0.2 ‰ for $\delta^{15}$N-NO$_3^-$ and 0.3 ‰ for $\delta^{18}$O-NO$_3^-$, which is within the same specification for $\delta^{15}$N-NO$_3^-$ and $\delta^{18}$O-NO$_3^-$ for at least duplicate measurements of the samples.

*Results*

**Page 6, section 3.1**

**RC:** It would be helpful to show T-S diagrams at this point rather than later in the discussion.

**AR:** We thought about the best position of the Sigma-theta-Salinity and Sigma-theta-Oxygen diagrams within our manuscript. In the end, we decided to show these diagrams with the distinct classification of the different water masses and the resultant water mass distribution model in a separate discussion section because of the high portion of discussion rather than just the presentation of results. In our water mass analyses, we use many different sources, describing water masses in the world's ocean and when available from expeditions in the Indian Ocean, but they are quite rare and no water mass model existed for our study area. Therefore, we decided to present the water mass distributions in an own discussion section and not as a part of the results. Consequently, the diagrams with the clear water mass classification along their density surfaces belong more to the discussion section. However, it would be a good opportunity to show a typical T-S diagram (see example below in addition to the salinity and oxygen color sections in Figures 2 a and b) in the results. These will give a first overview about the differences between northern and southern water masses and introduce the Figures and detailed explanation in the discussion part. This might be a good consensus.

[Figure]

Example for a T-S-diagram

**Page 7, lines 6-16 and Page 8, lines 2-13**

**RC:** Figure 5 (panels a, b, c, d) should be presented in this section and Table 2 moved to the supplementary materials. Figure 5 (panels e, and f) should be presented in this section and Table 3 moved to the supplementary materials.

**AR:** If we move Figure 5 to the results we will have to remove the overlay of water mass boundaries in the panels because they were added as a consequence of the water mass discussion section. Above we explained why we decided to present our water mass analyses as a part of the discussion. An opportunity to leave Figure 5 (a-f) in section 4.2.1 and to accommodate with your remarks is to add only nitrate and phosphate, and N and O isotope color sections (see example below; like Figure 2a and b for salinity and oxygen) to the results (3.2) and move Table 2 and 3 to the supplementary materials.

[Figure]

Example for nitrate, phosphate and N and O isotope transects

*Discussion*

**Page 9, line 6-7**

> **RC:** What is new in their water mass distribution model (Figure 4)?

> **AR:** This is the first water mass distribution model for this region, for further explanation see response above for "page 6, section 3.1"

**Page 9, lines 25-26**

> **RC:** Change for "… because of respiration and the absence of effective ventilation…"

> **AR:** We will rewrite the sentence as you mentioned.

**Page 9-12:**

> **RC:** It would be useful to include the nitrate isotopic composition (end-members) for the different water masses, either in Figure 4, or in a Table.

> **AR:** This is a good annotation. A type of endmembers are shown in Table 3 were the mean $\delta^{15}N$ and $\delta^{18}O$ are presented for different latitudes (because the water mass distribution changes along the transect) and for different water depth representing the different water masses. We can modify this Table and add the water masses for a better overview. Then we can move this Table to the beginning of section 4.2.1, after the water mass discussion part. It would be better to represent this new "end-member Table" for nitrate isotopes in section 4.2.1 rather than in section 3.1, because for the first time we connect the water masses and the results of the nutrient and isotopic measurements in section 4.2.1. This Table would than nicely correspond to Figure 5(a-f).

**Page 12, lines 20-21**

> **RC:** The $NO_3^-$/$PO_4^{3-}$ should however increase if N2-fixation is significant.

> **AR:** Enhanced N/P ratios in N-fixing organisms has been reported and would introduce these enhanced N/P ratios also to the water mass as the N-fixers are mineralised. This process is reflected in enhanced N/P ratios. The way we calculated the contribution from $N_2$-fixation is thus a minimum estimate of N contribution from $N_2$-fixation. If part of the P was also from N-fixers and if the N/P ratio of N fixers was known, their contribution could have been better estimated. However, we are not sure about the N-fixers N/P ratio. We will, however, examine this carefully in the revised version and improve this part including the dual isotope approach.

**Page 13, lines 10-11**

> **RC:** How does the mean $NO_3^-$/$PO_4^{3-}$ ratio changes along the latitudinal transect? What are the implications for N2 fixation?

> **AR:** The change of N/P ratio along the latitudinal transect is presented in Figure 5c and demonstrates the oligotrophic regime in the subtropical gyre. Implications for $N_2$-Fixation are: (1) Elevated N* values of >2 µM in surface

waters south of ~15°S. (2) We observe distinctly lower $\delta^{15}N$ values (<4.5-5.0 ‰) in the surface waters compared to the subsurface values within the SAMW with values of >7 ‰ and highest values of ~8 ‰. This leads to a $\delta^{15}N$ difference of 2-3.5 ‰, which is similar to the difference in other studies, i.e. Bourbonnais et al. (2009) with a difference of 3 ‰ (from 5 ‰ to 2 ‰). Our surface d15N values are also slightly lower than the average $\delta^{15}N$ values of depth water nitrate (>2000 m; 5.5 ‰). (3) We estimated in your simple calculation the input of new nitrate into the surface layer by $N_2$-fixation and demonstrated the increase of the N/P ratio of completely assimilated nitrate (Figure 9a).

**Page 14, line 1 and Page 18, line 16**

**RC:** Bourbonnais et al. (2009) is incorrectly referenced here.

**AR:** We apologize for the incorrectly referenced study. We will correct this.

**Page 15, lines 5-6**

**RC:** Add references to support this statement.

AR: We will add references.

**Page 16, lines 17-18**

**RC:** Why nitrate utilization is unlikely? It is too deep?

**AR:** It is to our knowledge a clear mixing signal that causes the moderate slopes of $\delta^{15}N/\ln(NO_3)$ in both the gyre region and in the Subantarctic and we think that nitrate utilization is unlikely at this depth. Sigman et al. (2000) also described the mixing of different end-members along the SAMW from the Antarctic with higher $\delta^{15}N$ (up to 13 ‰) values and lower $\delta^{15}N$ (<6 ‰, Liu et al., 1996) values towards lower latitudes.

**Page 18, lines 21-24**

**RC:** Bourbonnais et al. (2009) report a range of 2 to 5‰ for the δ15N of nitrate in surface waters of the subtropical northeast Atlantic Ocean. Using a simple isotopic mass balance, they estimated that N2 fixation could account for up to 40% of the export production in this region.

**AR:** This agrees with our $\delta^{15}N$ values, which are between 4.5 and 5.0 ‰ in surface waters and we estimated that $N_2$-fixation could account for ~30% of the export production. We will clarify this in the revised version.

**Page 18, lines 29-31**

**RC:** It is peculiar to note that the Δ(15,18) anomalies observed in this studies are at least half of the anomalies observed in the subtropical northeast Atlantic Ocean by Bourbonnais et al. (2009) (Δ(15,18) of -7 to 0‰). Why would that be if the estimated contribution from N2 fixation is supposedly in the same range (accounting for 30-40% of new supplied nitrate) for these two regions? The N* observed by Bourbonnais et al. (2009) was also up to ~3.5 µmol/kg.

**AR:** We will carefully examine this in the revised version using the suggested literature and include a discussion on the dual isotopes. We will reconsider the use of tracer Δ(15,18) because of the diverse source waters. Better would be the tracer Δ(15-18) from Rafter et al. (2013), who used only the difference between N and O isotope signatures which is more useful in regions characterised by a variety of water masses. We will consider this in the revised version.

**Page 19, lines 8-9**

**RC:** In equation (6), the nitrate to phosphate ratio (NO3-/PO43-) is divided by the measured phosphate concentrations, not multiplied.

**AR:** You are right, the equation is incorrect and N/Pcal must be multiplied by the phosphate concentrations. Sorry for this mistake.

**Page 19, lines 1-23**

**RC:** This approach requires many assumptions. One likely invalid assumption is assuming a Redfield ratio of 16. The Redfield ratio is variable in marine microalgae (see Geider et al., 2002). N2 fixers also have higher N/P ratios (e.g., Letelier et al., 1998). Finally, this approach does not take into account inputs from atmospheric depositions.

**AR:** Because $N_2$-fixers have higher N/P ratios, we calculated the assimilated nitrate by representing the deviation from the Redfield stoichiometry of 16:1 and therefore the higher N/P ratios of the assimilated nitrate are an evidence for $N_2$-fixation in surface waters (see comment above). We believe that we have presented a minimum estimate by our calculation but will re-examine our approach and try to find a better way to estimate the N-contribution by nitrogen fixers. We will check the literature on atmospheric deposition but we think that it is quite small in the study area as sinking particles and sediment shave only little lithogenic material.

**Page 19, line 21**

**RC:** This is confusing, as δ15N-NO3-fix (i.e. supplied from N2 fixation) should be about 0‰. I suggest removing the "fix" subscript.

**AR:** We agree with this remark, that $δ^{15}N\text{-}NO_3^-$fix is confusing; we will remove the subscript "fix".

**Page 19, lines 20-23**

**RC:** Overall, the dual nitrate isotopic data could be better exploited in their discussion and used in an isotopic box model to derive an independent assessment of the contribution from N2 fixation (see examples from Knapp et al., 2008 and Bourbonnais et al., 2009).

**AR:** Our simple estimation on $N_2$-fixation is the first try to get an impression on the input of new nitrate into the system of the subtropical gyre in the South Indian Ocean. For a box model we need to combine or water column analyses with the result of suspended matter samples and particle flux samples from sediment

traps. For this study, we first wanted to demonstrate the diversity of water masses in the less explored subtropical gyre of the South Indian Ocean and second, to highlight their varying influence on the nutrient and isotopic composition, which is likewise less investigated in this region. Our simple estimation on $N_2$-fixation is a first approach on the input of new nitrate into this special oligotrophic region. We will include the dual isotopes to strengthen our point on N-fixer contribution.

**Page 20, lines 9-10**

**RC:** Bourbonnais et al. (2009) did not observe significant positive Δ(15,18) anomalies in the subtropical northeast Atlantic Ocean. Which make me wonder what is the propagated (analytical) error associated with their Δ(15,18) measurements. In other words, is their calculated positive Δ(15,18) significantly different from 0?

**AR:** See above: We will consider to use the tracer Δ(15-18) instead of Δ(15,18) and agree that a Δ(15,18) of +0.5 or -0.5 ‰ is not a significant amplitude.

**Page 20, line 16**

**RC:** $N_2$-fixation have been shown to occur at lower temperatures in temperate regions (see Moisander et al., 2010).

**AR:** The sudden change in $\delta^{15}N$ and N* is difficult to explain in the gyre as nutrients are not increasing. We have no data on micronutrients but find it unlikely that these change significantly within the gyre. Therefore, the only feasible explanation seems to be the temperature drop. However, we will stress the contradictory literature in the revised version.

*Tables*

**Table 1**

**RC:** It is not necessary since the information is already presented in Figure 1. I recommend moving it to the supplementary materials.

**AR:** Table 1 will be moved to the supplement.

**Table 2 and 3**

**RC:** Should be moved to the supplementary materials as this information is already in Figures 3 and 5.

**AR:** See comment to Page 7, lines 6-16 and Page 8, lines 2-13

*Figures*

**Figure 1**

**RC:** It is difficult to see the shaded arrow representing the South equatorial current.

**AR:** We will highlight the shaded arrow by adding a contour line.

**Figure 6**

**RC:** What is the r2 and error on the slope?

**AR:** r2 is 0,99. We will add the r2 and the error of the slope in the revised version.

**Figure 7b**

**RC:** Which processes cause the positive $\Delta(15,18)$?

**AR:** See Sigman et al (2005): Nitrification/Remineralisation cycle in deeper waters leads to a slightly positive in $\Delta(15,18)$. We will add more information on that in the revised version.

*Technical comments*

**Page 1, lines 30-31 and Page 2, line 2**

**RC:** This sentence is repetitive. Replace by something like: "The South Indian Ocean is dominated by a subtropical anticyclonic gyre (refs), the Indian Ocean subtropical gyre" (IOSG), one of the major subtropical gyres in the world's ocean. The IOSG has been, thus far, sparsely investigated." Use the IOSG acronym defined earlier.

**AR:** We will rewrite the sentence as you mentioned and define the "IOSG" acronym earlier in the text.

**Page 3, line 10**

**RC:** Remove "Therefore" at the beginning of sentence.

**AR:** We will remove the "therefore".

**Page 12, line 7**

**RC:** Change for "nutrient distribution and N cycle processes"

**AR:** We will change the headline as you mentioned.

**Reply to the Review of the Referee #2 (Anonymous)**

(**RC**: Referee Comment; **AR**: Author's Responds)

First of all, thank you very much for reviewing our manuscript thoroughly and for the helpful comments and suggestions. We try to include as many comments and suggestions as possible which help us to improve our manuscript.

*Abstract*

**Page 1, line 20**

> **RC:** I would remove 'strong' here, as N* of -1 µM would not generally be considered "a strong N deficit".

> **AR:** We agree with you and have omitted "strong" here.

**Page 1, lines 21-23**

> **RC:** Please clarify what you are referring to here using "preformed versus regenerated". The preceding sentence referred to nitrate isotope signals coming from SAMW and from denitrification in the Arabian Sea. Where is the 'regenerated' signal that you are referring to?

> **AR:** The nitrate that is added by $N_2$-fixation and immediately consumed (assimilated) in the surface layer is meant as regenerated nitrate, as well as the remineralised nitrate by nitrification. These are in situ processes contrary to the preformed isotopic composition of nitrate induced by the lateral influence from the neighbouring water masses. We will rewrite the sentence to make this clearer.

**Page 1, lines 23-25**

> **RC:** If there is significant N2 fixation, I would not expect low nitrate to phosphate ratios. Revisit the N2 fixation discussion below.

> **AR:** Enhanced N/P ratios in N-fixing organisms has been reported and would introduce these enhanced N/P ratios also to the water mass as the N-fixers are mineralised. This process is reflected in enhanced N/P ratios. The way we calculated the contribution from $N_2$-fixation is thus a minimum estimate of N contribution from $N_2$-fixation. If part of the P was also from N-fixers and if the N/P ratio of N fixers was known, their contribution could have been better estimated. However, we cannot be sure about the N-fixers N/P ratio. We will, however, examine this carefully in the revised version and improve this part including the dual isotope approach.

*Introduction*

**Page 2, line 21**

> **RC: I** think a reference to Gruber and Sarmiento, 1997 would be appropriate here.

> **AR:** We will add the reference of Gruber and Sarmiento (1997).

**Page 2, lines 29-30**

> **RC:** The isotopic fractionation factor, $\varepsilon$, relates the instantaneous product, not the accumulated product, to the substrate. Though neither is explicitly stated, I think the implication is that this always holds true. This should be clarified.

> **AR:** We will examine this carefully and clarify this in our revised version.

*Materials and Methods*

**Page 6, line 14**

> **RC:** How is the 'single point correction' for $\delta15N$ applied? Is this simply a standard subtraction?

> **AR:** We will correct the method section, because we indeed do not use a single point correction but rather a two-point correction referred to IAEA-N3 ($\delta^{15}N\text{-}NO_3^-$ = +4.7 ‰ and $\delta^{18}O\text{-}NO_3^-$ = +25.6 ‰) and USGS-34 ($\delta^{15}N\text{-}NO_3^-$ = −1.8 ‰ and $\delta^{18}O\text{-}NO_3^-$ = −27.9 ‰) for $\delta^{15}N\text{-}NO_3^-$ and $\delta^{18}O\text{-}NO_3^-$.

*Results*

**Page 6, line 29**

> RC: What water mass does the 34.6 PSU feature represent?

> AR: AAIW (Antarctic Intermediate Water) is characterised by a salinity minimum of <34.6 PSU at a core density of 27.2 kg/m³. There is a mistake on page 10, line 2: "The salinity minimum (<34.9 PSU) south of 15° S …". We will correct this to "The salinity minimum (<34.6 PSU) south of 15° S…". Sorry for this misunderstanding.

**Figure 2**

> **RC:** It might be more helpful to include contours for the potential density surfaces, rather than contouring the same properties represented on the color bar.

> **AR:** This is a good remark and is a chance to include more information into the sections. We will change the couture lines for the potential density surfaces.

**Pages 7 and 8**

> **RC:**
> **(1)** I don't understand the choices behind what is shown in Tables 2 and 3. Why are these specific density/depth intervals selected, and why look at different density levels in the different latitude zones? Why are only one nitrate and

phosphate concentration (Table 2) or nitrate δ15N and δ18O value (Table 3) given for each line? How many measurements are included in these values? Shouldn't there be a range or uncertainty given for these if they derive from a range in latitude?

**(2)** Throughout this presentation of results in sections 3.2 and 3.3, I found referring to Figure 5 more useful than consulting Tables 2 and 3. I would suggest moving Figure 5 earlier in the paper, and removing Tables 2 and 3, or perhaps moving them to the supplement, unless their relevance can be better explained.

**AR:**
**(1)** For Table 2 and 3 we chose these depth intervals to provide average nitrate/phosphate concentrations (Table 2) and $\delta^{15}N/\delta^{18}O$ values (Table 3) for each water mass (along sigma-theta surfaces) in their specific latitudinal extent and thickness. Because the water mass distribution changes along the latitudinal transect, we chose 3 or 4 different latitudinal clusters to represent the change of nitrate/phosphate concentrations in Table 2 and isotopic compositions in Table 3. In Table 3 we decided to add a fourth latitudinal cluster to represent the divergent higher $\delta^{15}N$ and $\delta^{18}O$ values between 27.78°S and 26.05°S. For a better understanding of these Tables, it might be better to show the sigma-theta intervals rather than the depth intervals and to add the associated water masses to the density surfaces in the specific latitudinal section. We showed only one value for each latitude and depth range because these are averages of one to ten single values. You are right that we have to add a range for the average values. The uncertainties of each single measurement are shown in the supplement Table S1 and S2.

**(2)** If we move Figure 5 into the results we have to remove the overlay of water mass boundaries in the panels, because they were added as a consequence of the water mass discussion section. We thought intensely discussed this issue with all co-authors where the best position of the water mass section would be within our manuscript. In the end, we decided to present this section with the distinct classification of the different water masses and the resultant water mass distribution model in a separate discussion section because of the high portion of discussion rather than just the presentation of results. In our water mass analyses, we use many different sources, describing water masses in the world's ocean and when available from expedition in the Indian Ocean, but they are quite rare and no water mass model existed for our study area. Therefore, we decided to present the water mass distributions in an own discussion section and not as a part of the results. Consequently, we think that, Figure 5 with the overlying water mass distribution belongs to the discussion section. However, it would be a good opportunity to leave Figure 5 (a-f) in section 4.2.1, but add only nitrate, phosphate and nitrate isotope color sections without overlying water mass distributions (see example below; like Figure 2a and b for salinity and oxygen) to the results (3.2) and move the reworked Tables 2 and 3 to the supplementary materials.

[Figure]

Example for nitrate, phosphate and N and O isotope transects

**Page 7, line 6**

RC: I would delete 'strongly'. When working in oligotrophic areas, I'm not sure 5.9 µM nitrate qualifies as "strongly depleted". Otherwise, you could perhaps cite the concentration of nitrate in the surface waters, rather than at 310 m.

AR: This is a good objection. We will rewrite the sentence.

*Discussion*

**Page 10, line 27**

RC: Please clarify "decrease of the oxygen minimum". Do you mean that the oxygen concentration is increasing? If so, please rephrase.

AR: Yes, "decrease of the oxygen minimum", means that the oxygen concentration increases, we will rephrase the sentence.

**Figure 3**

RC: I didn't find this Figure necessary, and suggest that it be moved to the supplement.

AR: This Figure is intended to show how we defined the different water mass boundaries and how the water mass distribution model was generated. We think the presentation of these diagrams is very important for our water mass analyses and we would like to leave this Figure in the main text, unless you would necessarily move the Figure to the supplement.

**Figure 4**

RC: I think this Figure is extremely helpful for thinking about the water mass structure of the region! My only question is what determines where the lines dividing water masses are drawn? Are these specific sigma theta surfaces? Please clarify.

AR: To generate this water mass distribution model we use sigma-theta surfaces, and salinity and oxygen distribution (see Figure 3) to define the different water masses in the latitudinal transect. We separate the transect in

three latitudinal sections (see Figure 3): 20.36-27.78°S, 15.08°S and 2.98-8.81°S to represent the change of the water mass distribution along our transect. Therefore, we thought the presentation of Figure 3 helps to understand the generation of Figure 4.

**Figure 6**

**RC:** The Figure legend states that the color bar indicates potential density, but what is actually used is depth. Perhaps sigma theta would, in fact, be better.

**AR:** Sorry for this mistake! First, the color bar represents sigma-theta and was changed later into a depth color bar without a correction of the Figure legend. We will correct this and change the color bar for sigma-theta.

**Page 15, line 16**

**RC:** Doesn't iron availability also play a role in incomplete nitrate assimilation in the Southern Ocean?

**AR:** We will consider this and rewrite this paragraph.

**Page 17, line 3**

**RC:** Please clarify "lower water depths". Do you mean shallower or deeper?

**AR:** Sorry for the misunderstanding, we meant shallower water depths. We will make this clearer.

**Figure 8**

**RC:** The yellow star representing the mean nitrate d15N does not stand out. I would suggest making this symbol a different color or shape. Also, please provide the slope of the solid line in the Figure legend.

**AR:** We will highlight more clearly the symbol for the mean nitrate $\delta^{15}N$ and add the slope of the solid line in the Figure legend.

**Page 17, line 11**

**RC:** Typo, should be 'SAMW' rather than "SAWM".

**AR:** We will correct this mistake.

**Page 17, line 14**

**RC:** Please give Sigman et al., 2005 reference to $\Delta(15,18)$. Rafter et al (2013) is also a good reference, but uses $\Delta(15-18)$ instead.

**AR:** In our revised version be will reconsider the use of $\Delta(15,18)$ because of the diverse source waters in our study area. Better would be the tracer $\Delta(15-18)$ from Rafter et al. (2013), who used only the difference between N and O isotope signatures which is more useful in regions characterised by a variety of water masses. We will consider this in the revised version.

**Page 18, line 16**

**RC:** I would include a reference here to Gruber and Sarmiento 1997 for their seminal work in this area.

**AR:** We will include the reference to Gruber and Sarmiento (1997) here.

**Page 18, line 24**

**RC:** One could also reference work in the Atlantic from Knapp et al., 2008, and a variety of work from the Pacific.

**AR:** We will consider this and add more information from studies in the Atlantic and Pacific (Knapp et al.,2008 and Bourbonnais et al. 2009, etc.)

**Page 19, lines 7-8**

**RC:** What are the implications of assuming Redfield stoichiometry here?

**AC:** We will examine and clarify this in the revised version.

**Page 19, line 10**

**RC:** This equation appears incomplete, if not incorrect. From the text, I would not expect PO43-sample to appear in the denominator.

**AR:** Sorry for this mistake, you are right, the equation will be $NO_3^-cal = NO_3^-/PO_4^{3-}cal * PO_4^{3-}sample$

**Page 19, line 12**

**RC:** What is the N:P ratio assumed for newly fixed N? This seems important to the calculations performed here.

**AR:** We assumed an elevated N/P ratio for newly fixed N compared to the preformed N/P ratio in our study area (13.25 in the IOSG and 14.25 in the south equatorial Indian Ocean)**.** We observe these higher N/P ratios in our surface samples (low nitrate concentration of <5 µM) in Figure 9a.

**Page 19, lines 20-23**

**RC:** A newly fixed δ15N of 4.8‰ is not within the range of expected values for N2 fixation. This seems problematic, and requires reevaluation and justification of the approach used to arrive at this value. In my mind, a value of +4.8‰ argues against this N deriving from N2 fixation. What other explanations have the authors considered?

**AR:** Indeed, we found evidences for $N_2$-fixation: 1) Elevated N* values of >2 µM in surface waters south of ~15°S. (2) Even though a $\delta^{15}N$ of 4.5 to 5.0 % is not a clear evidence for the input of low $\delta^{15}N$ by $N_2$-fixation, the distinct decrease from subsurface values within the SAMW with values of >7 ‰ and highest values of ~8 ‰ indicate an upward decrease of 2-3.5 ‰. This decrease can only be explained by the input of fixed N into the surface layer. Additionally, studies in the Atlantic indicate the similar difference of 3 ‰ (from 5 ‰ to 2 ‰), i.e Bourbonnais et al. (2009). Furthermore the surface $\delta^{15}N$ values are lower than

the $\delta^{15}$N deep water mean of >5.5 ‰. (3) The calculated value of +4.8 ‰ is just a simple calculation there we take the calculated 30% of new N-input by N$_2$-fixation and compare this value with our observations assuming the SAMW as N source with a $\delta^{15}$N of >7 ‰. When the source would have higher $\delta^{15}$N values, than we would get higher $\delta^{15}$N values for the newly fixed $\delta^{15}$N in surface waters. In the revised version, we will make our arguments for N$_2$-fixation clearer.

**Figure 9**

**RC:** It is difficult to distinguish the symbols used to represent the two geographic areas in panel a. What calculation is used to derive the grey line in panel a?

**AR:** We will choose different symbols for a better differentiation in Figure 9a. The grey solid line indicates the calculated N-assimilation (regression line of NO$_3$-cal vs. NO$_3^-$/PO$_4^{3-}$cal) with a preformed NO$_3^-$/PO$_4^{3-}$in ratio of 13.25 for the region of the IOSG and 14.25 for the southern equatorial Indian Ocean and progressive nutrient assimilation with a Redfield ratio of 16 (NO$_3^-$/PO$_4^{3-}$ass).

**Page 20, line 16**

**RC:** Is low temperature the only other possible explanation? Increasing numbers of reports are finding N2 fixation at low temperature, thus the temperature limits seem to be a less convincing argument. What other contributing factors could be here?

**AR:** The sudden change in $\delta^{15}$N and N* is difficult to explain in the gyre as nutrients are not increasing. We have no data on micronutrients but find it unlikely that these change significantly within the gyre. Therefore, the only feasible explanation seems to be the temperature drop. However, we will stress the contradictory literature in the revised version.

**Page 21, lines 10-11**

**RC:** Can you make any connection here to the results of Martin and Casciotti, 2017 from the Arabian Sea?

**AR:** We will consider to add some sentences with respect to Martin and Casciotti (2017), but their work focuses on the Arabian Sea and not on the influence from the Arabian Sea on the south Indian Ocean.

**Changes in the revised manuscript: "Nutrient distribution and nitrogen and oxygen isotopic composition of nitrate in water masses of the subtropical South Indian Ocean"**

by Natalie C. Harms[1], Niko Lahajnar[1], Birgit Gaye[1], Tim Rixen[1,2], Kirstin Dähnke[3], Markus Ankele[3], Ulrich Schwarz-Schampera[4] and Kay-Christian Emeis[1,3]

[1]Institute of Geology, Universität Hamburg, Hamburg, 20146, Germany.
[2]Leibniz-Centre for Tropical Marine Research, Bremen, 28359, Germany.
[3]Helmholtz-Zentrum Geesthacht (HZG), Institute for Coastal Research, Geesthacht, 21502, Germany.
[4]Federal Institute for Geosciences and Natural Resources (BGR), Hannover, 30655, Germany.

Pages and lines refer to the marked-up manuscript below.

**Abstract**

Page 1, line 10 - Page 2, line 8

> We rewrote the Abstract in order to clarify our new findings (new nutrient and isotope data in a less explored ocean region accompanied by the first water mass distribution model in this ocean region) and added the depth range of the N* minimum. Furthermore, we added a sentence to the supply of newly fixed nitrogen in surface waters of the Indian Ocean subtropical gyre.

**Introduction**

Page 3, lines 2-3 and 5-6

> We modified the sentences as Referee #1 suggested.

Page 3, lines 17-19

> We modified the sentence.

Page 3, line 27

> We added the reference Gruber and Sarmiento (1997).

Page 4, line 1-4

> We added a paragraph to demonstrate the limitation of the tracer N*, and to strengthen the importance of using stable isotopes to overcome the weakness associated with the N* approach, following by the comments of Referee #1.

Page 4, lines 7-14

> We shifted the formula for the δ-notation of nitrate $\delta^{15}N$ and $\delta^{18}O$ from the "Materials and Methods" section (Page 8, lines 30-31) to the "Introduction" section and added the definition of the isotope effect ε.

Page 4, lines 20-29:

> We rewrote this paragraph to explain in more detail the benefit from the decoupling effect of N and O isotopes in nitrate regarding consumption and production processes. Additionally, we deleted tracer Δ(15,18) which is replaced by Δ(15-18) mentioned by Rafter et al., (2013) and is noted later in the manuscript.

Page 5, lines 11-13:

We added a sentence to note the lateral influence of preformed nutrients from the neighbouring ocean regions.

**Materials and Methods**

We shifted Table 1 (CTD station list from the previous manuscript) to the supplement material.

Figure 1

To highlight the grey arrow that marks the South Equatorial Current (SEC), we added a black contour line to Figure 1.

Page 8, lines 22-23

We corrected the description of the measurement method of N and O isotopes, because we indeed use a two-point correction referred to IAEA and USGS standards.

Page 8, lines 30-31

We shifted the formula for the δ-notation of nitrate $\delta^{15}N$ and $\delta^{18}O$ from the "Materials and Methods" section to the "Introduction" section (Page 4, lines 7-9).

**Results**

Figure 2

In both Figures, 2a and 2b, black contour lines were changed into density surfaces, following the suggestion of Referee #2.

Figure 3

We added a temperature-salinity diagram, accompanied with oxygen concentrations, to demonstrate the clear difference of water masses in the southern and northern study area and to introduce Figure 5, presented later in the manuscript.

Page 10, line 10 - Page 11, line 6

We demonstrate more clearly the nutrient depleted surface layer and corrected the nutrient values mentioned in the density range of 24.9−26.4 kg/m³ (<300 m). Furthermore, we added the nutrient concentration in the depth range of the oxygen maximum (σ= 26.4−26.9 kg/m³).

Figure 4 and Table 1 (revised manuscript)

We added Figure 4 to replace Tables 2 and 3 from the previous manuscript. These Tables are combined to one Table that now is Table 1 in the revised manuscript. Additionally, a column for water masses was added to get a better overview on the nutrient distribution and isotopic composition of nitrate in the individual water masses, which are explained in the following section. Therefore, we followed the suggestion of the Referee #1 and Referee #2.

**Discussion**

Page 14, lines 8-11

We added sentences to explain the development of the water mass distribution model by using the density-salinity and density-oxygen diagrams in Figure 5 and strengthen the importance of Figure 5.

Page 15, lines 5-11

We added more information on the Subantractic Mode Water (SAMW) based on the study of Herraiz-Borreguero and Rintoul (2011), as suggested by the editor.

Page 15, line 11-13

We modified this sentence referred to the suggestion of Referee #1.

Page 15, line 17 and Page 16, line 5

We corrected the general density range of the boundary between intermediate and deep water masses.

Page 16, line 10-13

We modified the sentence to clarify that a further increase in oxygen as well as salinity at the 2°C level mark the transition from the Indian Deep Water (IDW) to the underlying Circumpolar Deep Water (CDW).

Page 20, line 1

We modified the subheading 4.2 as Referee #1 mentioned.

Page 20, lines 10-13

We added concrete values to demonstrate the increase in nutrient concentrations within the Indian Equatorial Water (IEW).

Figure 7

We added a latitudinal section (Figure 7g) to demonstrate the decoupling of N and O isotopes in our study area, especially within the Subantarctic Mode Water (SAMW) and the Red Sea-Persian Gulf Intermediate Water (RSPGIW). Therefore we used the tracer $\Delta(15-18)$ instead of $\Delta(15,18)$.

Page 23, line 7 and Page 31 line, 16

We removed the reference Bourbonnais et al. (2009), as required by Referee #1.

Page 23, lines 8-9

We added a sentence to the analytical error of N* values.

Figure 8

We corrected the colour-coding of the sample dots from "depth" to "density" and added the r² of the slope.

Page 25, lines 6-14

We rewrote this paragraph and added $\delta^{15}N$ values from the literature, detected in the Arabian Sea.

Page 25, line 15 - Page 26, line 12

We added a paragraph about the decoupling effect of N and O isotopes and the resulting elevated $\Delta(15-18)$ values in the Red Sea-Persian Gulf Intermediate Water (RSPGIW) and in the lower Indian Equatorial Water (IEW).

Page 26, lines 23-24

We rewrote these sentences and mentioned that also reduced iron availability could lead to partial N-assimilation in the Subantarctic region as well as light limitation.

Figure 9

In the previous manuscript, we removed Figure 7b, because we replaced tracer Δ(15,18) by tracer Δ(15-18) that is presented as latitudinal section in Figure 7g in the revised version. Additionally, we shifted Figure 8 from the previous manuscript to Figure 9 in the revised manuscript, which it is now Figure 9b. Furthermore, in Figure 9b, we changed the colour signatures for the different water masses consistently with the water mass classification in Figure 9a and added the Figure legend, following by the suggestions of Referee #2.

Page 30, line 9 - Page 31, line 8

We added sentences to the decoupling effect of N and O isotopes in Subantractic Mode Water (SAMW) leading to lower Δ(15-18) values.

Page 31, line 15

We added the reference Gruber and Sarmiento (1997) and removed the reference to Bourbonnais et al. (2009), as required by both Referees.

Page 31, lines 17-23

Following by the suggestion of Referee #1, we rewrote this paragraph and added some information on the limitation of the N* approach and on atmospheric deposition and why we neglect this point in our further discussion.

Page 31 line 27 - Page 32, line 6

We rewrote this paragraph to make clear the increase of $\delta^{15}N$ values towards the surface layer (upper 200 m) compared to the underlying SAMW with strong elevated $\delta^{15}N$ values. We compared this upward increase accompanied with a lower Δ(15-18) in surface waters with studies in the NW Atlantic, indicating a similar upward increase, caused by $N_2$-fixation.

Page 32, lines 23-24

We corrected the formula (2).

Page 33, line 4-14

We corrected the mean $\delta^{15}N$ of the SAMW (7.4 ‰ instead of 7.3 ‰). For the mean $\delta^{15}N$ of the surface water we take only into account the upper 200 m at latitude 20.36° S−23.91° S (5.0 ‰), which results in a calculated $\delta^{15}N\text{-}NO_3^-{}_{new}$ of ~4.9 ‰. We compared our results with the literature and backed our assumption of $N_2$-fixation in surface waters of the ISOG at 20.36° S−23.91° S.

Figure 10

In Figure 10a, we modified colour and shape of the coloured dots that represent samples at latitudes 26.06° S-27.78° S, following by the comment of Referee #2. Additionally, instead of "$NO_3^-{}_{fix}$" we named the supply of newly fixed nitrogen to the nitrate pool "$NO_3^-{}_{new}$". See also in the text at Page 32, line 29 - Page 33, line 11.

Page 35, lines 7-27

Following the suggestions of both Referees, we rewrote this paragraph and added the aspect of reduced iron availability as further explanation for the abrupt decline of $N_2$-fixation south of ~26° S.

**Conclusion**

Page 36, lines 16-18

We adjusted this sentence according to the new tracer $\Delta$(15-18).

Page 37, lines 2-9

We added information to the data availability of water column parameters, and nutrient and isotope data from research cruises MSM 59/2 in 2016 and SO 259 in 2017.

**References**

We adjusted the references according to our changes in the manuscript.

**Formatvorlagendefinition:** Absatz-Standardschriftart

[revised manuscript text omitted]

---

## Referee Report (RR1)

**Review of Harms *et al.* (iteration #2)**

**General comments:**
Overall, Harms *et al.* addressed most of my concerns. The dataset presented is interesting as it focuses on an under-sampled region of the ocean (the subtropical South Indian Ocean). While they now use their dual nitrate isotope data a bit more in their discussion, I was not entirely satisfied. In my opinion, this section of the manuscript remains insufficient. The study would be improved if they could use their dual nitrate isotope data in a simple model to validate their current estimate of newly fixed N derived using Redfield assumptions. I have provided the equations that I use for similar simple isotope box models in my 2009 and 2013 manuscripts below. The authors are welcome to contact me (abourbonnais@seoe.sc.edu) with any questions.

**Specific comments and technical corrections:**

**Abstract:**
Page 1, lines 14-15: Change to: "Our results are the first in this ocean region and provide new information on nitrogen sources and transformation processes."

Page 1, line 16: Change to "... IOSG with values of $<3$ µM for both $NO_3^-$ and $PO_4^{3-}$...."

page 1, lines 23-24: What do they mean by "partial-assimilated organic matter"? Do they mean organic matter with a low $\delta^{15}N$? Is there any evidence for low $\delta^{15}N$ of organic material in the SAMW?

Page 1, line: They did not use an isotope budget to derive this estimate (see my general comments above).

**Materials and Methods:**

page 5, lines 19-20: What was the $\delta^{15}N$ and $\delta^{18}O$ for the internal standard? What was the size of the blank (if any)?

**Results:**

Page 7, line 9: Change to ".... of $<3$ µM for both $NO_3^-$ and $PO_4^{3-}$"

**Discussion:**

Page 14, lines 2-3: Change to "and biologically $N_2$-fixation are major processes..."

Page 15, lines 9-11: Change to "... N*. The analytical error on N* estimate based on the relative error for nitrate and phosphate analysis was below 1.5% for duplicate sample measurements."
Is the error on N* calculated from propagating the errors for these analysis?

page 15, line 15: Change "characterized" to "affected".

Page 16, lines 11-13: $^{18}\varepsilon:^{15}\varepsilon$ is associated with both assimilative (nitrate assimilation) and dissimilative (denitrification) nitrate reduction.

Page 16, lines 15-17: This sentence is not clear. The isotope effect associated with ammonium production and nitrification does not affect the $\delta^{15}N\text{-}NO_3^-$ because ammonium and nitrite generally does not accumulate in oxic waters.

Page 16 and 17, last and first sentences: This point needs to be discussed more. How the $\delta^{18}O$ of seawater is affected by ammonium and nitrite oxidation? For instance, they could add the following sentence: "$\delta^{18}O$ depends on the $\varepsilon$ during $NH_4^+$ and $NO_2^-$ oxidation, water incorporation (with $\delta^{18}O\text{-} H_2O$ of ~0‰), and the exchange of oxygen atoms with water that should generate a $\delta^{18}O$ of newly produced $NO_3^-$ between -8 and -1‰ (Buchwald and Casciotti, 2010; Casciotti *et al.*, 2010).

Page 19, lines 13-14: Provide an estimate of N atmospheric depositions in this region to support this claim.

Page 19, line 22: Correct "N₂-fiaxtion"

Page 19, section 4.2.2. The manuscript would be improved if they could also derive an N₂ fixation estimate using a simple isotope model, as described below.

**Additional references:**

Bourbonnais, A., Lehmann, M. F., Hamme, R. C., Manning, C. C., & Juniper, S. K. (2013). Nitrate elimination and regeneration as evidenced by dissolved inorganic nitrogen isotopes in Saanich Inlet, a seasonally anoxic fjord. *Marine chemistry*, *157*, 194-207.

Casciotti, K. L., McIlvin, M., & Buchwald, C. (2010). Oxygen isotopic exchange and fractionation during bacterial ammonia oxidation. *Limnology and Oceanography*, *55*(2), 753-762.

**Example of simple isotope box model (Bourbonnais *et al.,* 2009 and 2013)**

[Figure]

**Figure 1.** Nitrate isotope box-model for scenario 2a in Bourbonnais *et al.* (2013).

**Note: $\delta^{15}N_{rain}$ in this model represents the $\delta^{15}N$ in precipitations, but could be replaced by the $\delta^{15}N$ for $N_2$ fixation in the Harms *et al.* paper.**

N flux terms:

**a** = N from precipitation flux

**b** = seawater mixing

**c** = nitrate removal (by nitrate assimilation ($C_{upt}$)

**d** = organic matter remineralization and nitrification (recycled production)

$\delta^{15}N_{Box}$ = steady-state $\delta^{15}N\text{-}NO_3^-$ (under model assumptions)

$\delta^{18}O_{Box}$ = steady-state $\delta^{18}O\text{-}NO_3^-$ (under model assumptions)

$\delta^{15}N_{rain}$ = $\delta^{15}N\text{-}NO_3^-$ from precipitation (2‰)

$\delta^{18}O_{rain}$ = $\delta^{18}O\text{-}NO_3^-$ from precipitation (35‰)

$\delta^{15}N_{dw\ mix}$ = $\delta^{15}N\text{-}NO_3^-$ added from vertical mixing with deep-water (7.5‰)

$\delta^{18}O_{dw\ mix}$ = $\delta^{18}O\text{-}NO_3^-$ added from vertical mixing with deep-water (2‰)

$\delta^{15}N_{upt}$ = $\delta^{15}N_{Box}$ - $\varepsilon_{upt}$ (‰) = $\delta^{15}N\text{-}NO_3^-$ of nitrate assimilated

$\delta^{18}O_{upt}$ = $\delta^{18}O_{Box}$ - $\varepsilon_{upt}$ (‰) = $\delta^{18}O\text{-}NO_3^-$ of nitrate assimilated

where

$\varepsilon_{upt}$ (‰) = nitrate assimilation isotope effect (5 ‰)

$\delta^{15}N_{nitrif}$ = $\delta^{15}N$ of nitrate generated in the process of organic matter remineralization and nitrification.

$\delta^{18}O_{nitrif}$ = $\delta^{18}O$ of nitrate generated in the process of organic matter remineralization and nitrification (-3.8 ‰).

The equation for $\delta^{15}N_{Box}$ is derived from:

$\delta^{15}N_{Box} = [\delta^{15}N_{rain} \times a] + [\delta^{15}N_{dw\ mix} \times b] - [(c_{upt} - d) \times (\delta^{15}N_{Box} - {}^{15}\varepsilon_{upt})]$

$\delta^{15}N_{Box} = [\delta^{15}N_{rain} \times a] + [\delta^{15}N_{dw\ mix} \times b] - [(c_{upt} - d) \times \delta^{15}N_{Box}] + [(c_{upt} - d) \times {}^{15}\varepsilon_{upt}]$

$\delta^{15}N_{Box} + [(c_{upt} - d) \times \delta^{15}N_{Box}] = [\delta^{15}N_{rain} \times a] + [\delta^{15}N_{dw\ mix} \times b] + [(c_{upt} - d) \times {}^{15}\varepsilon_{upt}]$

$\delta^{15}N_{Box} + [(c_{upt} - d) \times \delta^{15}N_{Box}] = [\delta^{15}N_{rain} \times a] + [\delta^{15}N_{dw\ mix} \times b] + [(c_{upt} - d) \times {}^{15}\varepsilon_{upt}]$

$\delta^{15}N_{Box} \times [1 + (c_{upt} - d)] = [\delta^{15}N_{rain} \times a] + [\delta^{15}N_{dw\ mix} \times b] + [(c_{upt} - d) \times {}^{15}\varepsilon_{upt}]$

The final equation for $\delta^{15}N_{Box}$ is:

$$\delta^{15}N_{Box} = [(\delta^{15}N_{rain} \times a] + [\delta^{15}N_{dw\ mix} \times b] + [(c_{upt} - d) \times {}^{15}\varepsilon_{upt}] / [1 + (c_{upt} - d)]$$

The equation for $\delta^{18}O_{box}$ is derived from:

$$\delta^{18}O_{Box} = [\delta^{18}O_{rain} \times a] + [\delta^{18}O_{sw\ mix} \times b] - [c_{upt} \times (\delta^{18}O_{Box} - {}^{18}\varepsilon_{upt})] + [d \times \delta^{18}O_{nitrif}]$$

$$\delta^{18}O_{Box} = [\delta^{18}O_{rain} \times a] + [\delta^{18}O_{sw\ mix} \times b] - [c_{upt} \times \delta^{18}O_{Box}] + [c_{upt} \times {}^{18}\varepsilon_{upt})] + [d \times \delta^{18}O_{nitrif}]$$

$$\delta^{18}O_{Box} + [c_{upt} \times \delta^{18}O_{Box}] = [\delta^{18}O_{rain} \times a] + [\delta^{18}O_{sw\ mix} \times b] + [c_{upt} \times {}^{18}\varepsilon_{upt})] + [d \times \delta^{18}O_{nitrif}]$$

$$\delta^{18}O_{Box} \times [1 + c_{upt}] = [\delta^{18}O_{rain} \times a] + [\delta^{18}O_{sw\ mix} \times b] + [c_{upt} \times {}^{18}\varepsilon_{upt})] + [d \times \delta^{18}O_{nitrif}]$$

The final equation for $\delta^{18}O_{Box}$ is:

$$\delta^{18}O_{Box} = [\delta^{18}O_{rain} \times a] + [\delta^{18}O_{sw\ mix} \times b] + [c_{upt} \times {}^{18}\varepsilon_{upt})] + [d \times \delta^{18}O_{nitrif}] / [1 + c_{upt}]$$

The $\Delta(15,18)$ is calculated according to the following equation:

$$\Delta(15,18) = (\delta^{15}N - \delta^{15}N_{deep}) - [({}^{18}\delta/{}^{15}\delta) \ (\delta^{18}O - \delta^{18}O_{deep})]$$

---

## Author Response (AR2)

**Reply to the Reviews of Referee #1 (Bourbonnais) and Referee #2 (Anonymous) including the relevant changes in the revised manuscript: "Nutrient distribution and nitrogen and oxygen isotopic composition of nitrate in water masses of the subtropical South Indian Ocean"**

**Reply to the Review of Referee #1 (Bourbonnais)**

(**RC#1**: Referee #1 Comment; **AR**: Author's Responds; Page and line numbers in brackets according to the marked-up manuscript below)

First of all, thank you again very much for thoroughly reviewing our revised manuscript and for the helpful comments and suggestions. We included as many comments and suggestions as possible which helped us to improve our manuscript.

*Abstract*

**Page 1, lines 14-15 (Page 1, lines 14-15):**

> **RC#1:** Change to: "Our results are the first in this ocean region and provide new information on nitrogen sources and transformation processes."

> **AR:** We changed the sentence as you suggested.

**Page 1, line 16 (Page 1, lines 15-17):**

> **RC#1:** Change to "…IOSG with values of >3 µM for both $NO_3^-$ and $PO_4^{3-}$…"

> **AR:** We restructured this sentence.

**Page 1, lines 23-24 (Page 1, lines 24-26):**

> **RC#1:** What do they mean by "partial-assimilated organic matter"? Do they mean organic matter with a low $\delta^{15}N$? Is there any evidence for low $\delta^{15}N$ of organic material in the SAMW?

> **AR:** "Partial-assimilated organic matter" is organic matter that is produced during particle N-assimilation in the Subantarctic region. Partial N-assimilation has a distinct isotope effect and therefore produce organic matter that is depleted in $^{15}N$, while the residual nitrate pool, in this case the SAMW, is enhanced in $^{15}N$. This leads to elevated $\delta^{15}N$ values that are observed within the SAMW. Because in this manuscript we do not present suspended matter or particle flux data, we are not able to present $\delta^{15}N$ values of particulate organic matter within the SAMW. However, the strongly enhanced $\delta^{15}N$ values within the ambient water is clear evidence for partial N-assimilation coming from the Subantarctic region via the SAMW.

*Materials and Methods*

**Page 5, lines 19-20 (Page 6, lines 19-20):**

> **RC#1:** What are the $\delta^{15}N$ and $\delta^{18}O$ for the internal standard? What was the size of the blank?

> **AR:** The internal standard KBI has a $\delta^{15}N\text{-}NO_3^-$ of 7.1 ‰. We added this information. Blank size was in most of the measurements below detection limit.

*Results*

**Page 7, line 9 (Page 8, lines 9-10):**

> **RC#1:** Change to "… of <3 µM for both $NO_3^-$ and $PO_4^{3-}$".

> **AR:** We rephrased the sentence.

*Discussion*

**Page 14, line 2-3 (Page 17, lines 13-14):**

> **RC#1:** Change to "and biologically $N_2$-fixation are major processes…".

> **AR:** We rephrased the sentence as Referee #1 mentioned.

**Page 15, lines 9-11 (Page 20, lines 5-7):**

> **RC#1:** Change to "…N*. The analytical error on N* estimate based on the relative error for nitrate and phosphate analysis was below 1.5 % for duplicate sample measurements." Is the error on N* calculated from propagating the errors for these analysis?

> **AR:** We rephrased this sentence as Referee #1 mentioned. The error on N* was not calculated from propagating the errors of nitrate and phosphate analysis.

**Page 15, line 15 (Page 20, line 12):**

> **RC#1:** Change "characterized" to "affected".

> **AR:** We changed the wording.

**Page 16, lines 11-13 (Page 22, lines 11-12):**

> **RC#1:** $^{18}\varepsilon{:}^{15}\varepsilon$ is associated with both assimilative (nitrate assimilation) and dissimilative (denitrification) nitrate reduction.

> **AR:** We agree with this statement. We wrote this in the introduction but in this context, we want to highlight the effects related to denitrification.

**Page 16, lines 15-17 (Page 22, lines 15-17):**

> **RC#1:** This sentence is not clear. The isotope effect associated with ammonium production and nitrification does not affect the $\delta^{15}N\text{-}NO_3^-$ because ammonium and nitrite generally does not accumulate in oxic waters.

> **AR:** We wrote that "…the N isotope effect associated with ammonium production and nitrification does not affect the $\delta^{15}N\text{-}NO_3^-$ but depends on the biomass being remineralised…".

**Page 16 and 17, last and first sentence (Page 22, lines 18-20):**

> **RC#1:** This point needs to be discussed more. How the $\delta^{18}O$ of seawater is affected by ammonium and nitrite oxidation? For instance, they could add the following sentence: "$\delta^{18}O$ depends on the $\varepsilon$ during $NH_4^+$ and $NO_2^-$ oxidation, water incorporation (with $\delta^{18}O\text{-}H_2O$ of ~0 ‰), and the exchange of oxygen atoms with water that should generate a $\delta^{18}O$ of newly produced $NO_3^-$ between -8 and -1 ‰ (Buchwald and Casciotti, 2010; Casciotti et al., 2010).

> **AR:** We added the information on processes that can modify the $\delta^{18}O$ as Referee #1 suggested.

**Page 19, lines 13-14 (Page 25, lines 24-25):**

> **RC#1:** Provide an estimate of N atmospheric depositions in this region to support this claim.

> **AR:** Until now there is no solid information on the quantity of atmospheric deposition in this ocean region, therefore we are very careful with exact statements. However, it is mentioned that the input of N by atmospheric deposition is one of the lowest compared to other ocean regions (Duce at al., 2008; Duce and Tindale, 1991).

**Page 19, section 4.2.2. (Page 25-27, section 4.2.2.):**

**RC#1:** The manuscript would be improved if they could also derive an $N_2$ fixation estimate using a simple isotope model, as described below.

**AR:** We thought about adding this isotope box model and had a detailed look on the equations, which were outlined in the Review of Referee #1. We decided not to add the complete box model estimation, but we took the final equation for $\delta^{15}N_{box}$ and used this to rewrite and prove our N/P ratio-based calculation estimating the amount of N that is added by $N_2$-fixation in surface waters of the IOSG. Because N-assimilation in oligotrophic environments has no net expression of the isotope fractionation and the remineralisation of the produced organic matter has therefore most likely no concrete isotope effect, we can neglect the second part of the box model equation for $\delta^{15}N$. As written in the section 4.2.2., the percentage of 32 % from box model estimation calculated for the amount of N that is added to the nitrate pool by $N_2$-fixation agrees quite well with the 34 % calculated by our N/P ratio estimation.

**Reply to the Review of Referee #2 (Anonymous)**

(**RC#2**: Referee #2 Comment; **AR**: Author's Responds; Page and line numbers in brackets according to the marked-up manuscript below)

First of all, thank you again very much for thoroughly reviewing our revised manuscript and for the helpful comments and suggestions. We included as many comments and suggestions as possible which helped us to improve our manuscript.

**Introduction**

**Page 2, line 28 (Page 3, line 28):**

> **RC#2:** This equation for N*, including the constant 2.9 is in units of µmol/kg.

> **AR:** µmol/kg is often written as µM for the concentration of nitrate and phosphate and therefore also for N* (see e.g. Martin and Casciotti, 2017). Nevertheless, to prohibit confusion we changed the concentration of nitrate, phosphate and N* into the unit µmol/kg. According to these changes, we corrected the units within the Figures 4, 7, 8, 9 and 10, as well as in Table 1.

**Page 3, line 15 and Page 16, line 16 (Page 4, line 15 and Page 22, line 16):**

> **RC#2:** Typo on $\delta^{15}N\text{-}NO_3^-$.

> **AR:** We corrected this.

**Discussion**

**Page 16, line 8 (Page 22, line 6-8):**

> **RC#2:** You could also compare your results to Martin and Casciotti, 2017: Paired N and O isotope analysis of nitrate and nitrite in the Arabian Sea oxygen deficit zone.

> **AR:** We added the reference to Martin and Casciotti (2017), which agree with the findings of Gaye et al. (2013), regarding $\delta^{15}N$ and $\delta^{15}O$ values in the oxygen minimum zone within the Arabian Sea.

**Page 17, lines 4-6 (Page 22, lines 25-28):**

> **RC#2:** This is certainly one possibility. In this region, I think the authors should also discuss the potential for oxygen minimum zone processes to alter Δ(15-18), since this is an idea that is out there in the literature. If they don't think that is important here, they could say so, but I think some mention is warrented.

> **AR:** We added two sentences to mention this point.

**Page 18, lines 13-16 (Page 25, lines 9-12):**

> **RC#2:** This sentence construction doesn't quite make logical sense. I would suggest clarification.

> **AR:** We restructured these sentences.

**Page 19, line 13 (Page 25, line 27):**

> **RC#2:** Should this be "…the influx of nutrient enriched mineral aerosols…"?

> **AR:** Sorry for this confusing scribal error.

**Page 19, line 14 (Page 25, line 28):**

> **RC#2:** I'd suggest "…neglect this factor in our further discussion."

> **AR:** We corrected this.

**Page 20 (Page 26-27, section 4.2.2):**

**RC#2:** I found this whole section somewhat confusing, and potentially problematic, starting with equation 1. I think this section would be much clearer if they started with an N mass balance, which includes uptake and remineralisation. Perhaps in your P mass balance, you can bring in the N/P ratios, but using if from the start is extremely confusing. It's not obvious in these equations where the remineralised N and P are, through that is where the N2 fixation signal should reside.

**AR:** We restructured this section and added the equation of the box model estimation mentioned in the Review of Referee #1 to strengthen our calculation using the N/P ratio. (For details, see the reply to Referee #1 above and section 4.2.2).

**Formatvorlagendefinition:** Absatz-Standardschriftart

[revised manuscript text omitted]